# The Memory Perturbation Equation: Understanding Model's Sensitivity to Data

**Peter Nickl**[†]
peter.nickl@riken.jp

**Lu Xu**[*†]
lu.xu.sw@riken.jp

**Dharmesh Tailor**[*‡]
d.v.tailor@uva.nl

**Thomas Möllenhoff**[†]
thomas.moellenhoff@riken.jp

**Mohammad Emtiyaz Khan**[†§]
emtiyaz.khan@riken.jp

## Abstract

Understanding model's sensitivity to its training data is crucial but can also be challenging and costly, especially during training. To simplify such issues, we present the Memory-Perturbation Equation (MPE) which relates model's sensitivity to perturbation in its training data. Derived using Bayesian principles, the MPE unifies existing sensitivity measures, generalizes them to a wide-variety of models and algorithms, and unravels useful properties regarding sensitivities. Our empirical results show that sensitivity estimates obtained during training can be used to faithfully predict generalization on unseen test data. The proposed equation is expected to be useful for future research on robust and adaptive learning.

## 1 Introduction

Understanding model's sensitivity to training data is important to handle issues related to quality, privacy, and security. For example, we can use it to understand (i) the effect of errors and biases in the data; (ii) model's dependence on private information to avoid data leakage; (iii) model's weakness to malicious manipulations. Despite their importance, sensitivity properties of machine learning (ML) models are not well understood in general. Sensitivity is often studied through empirical investigations, but conclusions drawn this way do not always generalize across models or algorithms. Such studies are also costly, sometimes requiring thousands of GPUs [38], which can quickly become infeasible if we need to repeat them every time the model is updated.

A cheaper solution is to use local perturbation methods [21], for instance, influence measures that study sensitivity of trained model to data removal (Fig. 1(a)) [8, 7]. Such methods too fall short of providing a clear understanding of sensitivity properties for generic cases. For instance, influence measures are useful to study trained models but are not suited to analyze training trajectories [14, 54]. Another challenge is in handling non-differentiable loss functions or discrete parameter spaces where a natural choice of perturbation mechanisms may not always be clear [32]. The measures also do not directly reveal the causes of sensitivities for generic ML models and algorithms.

In this paper, we simplify these issues by proposing a new method to unify, generalize, and understand perturbation methods for sensitivity analysis. We present the Memory-Perturbation Equation (MPE) as a unifying equation to understand sensitivity properties of generic ML algorithms. The equation builds upon the Bayesian learning rule (BLR) [28] which unifies many popular algorithms

---

[*]Equal contribution. Part of this work was carried out when Dharmesh Tailor was at RIKEN AIP.

[†]RIKEN Center for AI Project, Tokyo, Japan.

[‡]University of Amsterdam, Amsterdam, Netherlands.

[§]Corresponding author.

37th Conference on Neural Information Processing Systems (NeurIPS 2023).

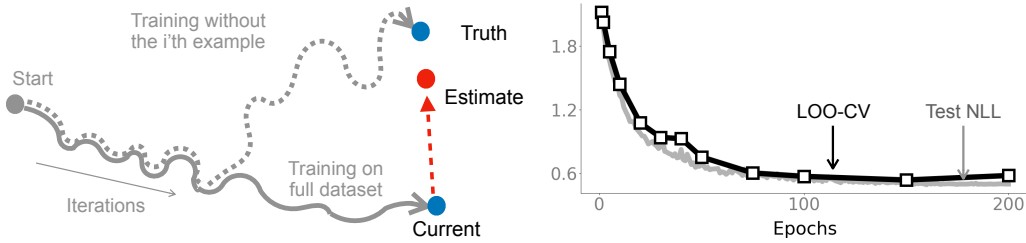

(a) Estimating the effect of an example removal     (b) Predicting test NLL during training, CIFAR10

Figure 1: Our main goal is to estimate the sensitivity of the training trajectory when examples are perturbed or simply removed; see Panel (a). We present the MPE to estimate the sensitivity without any retraining and use them to faithfully predict the test performance from training data alone; see Panel (b). The test negative log-likelihood (gray line) for ResNet–20 on CIFAR10 shows similar trends to the leave-one-out (LOO) score computed on the training data (black line).

from various fields as specific instances of a *natural-gradient* descent to solve a Bayesian learning problem. The MPE uses natural-gradients to understand sensitivity of all such algorithms. We use the MPE to show several new results regarding sensitivity of generic ML algorithms:

1. We show that sensitivity to a group of examples can be estimated by simply adding their natural-gradients; see Eq. 6. Larger natural-gradients imply higher sensitivity and just a few such examples can often account for most of the sensitivity. Such examples can be used to characterize the model's memory and memory-perturbation refers to the fact that the model can forget its essential knowledge when those examples are perturbed heavily.

2. We derive Influence Function [8, 31] as a special case of the MPE when natural-gradients with respect to Gaussian posterior are used. More importantly, we derive new measures that, unlike influence functions, can be applied *during* training for all algorithms covered under the BLR (such as those used in deep learning and optimization). See Table 1.

3. Measures derived using Gaussian posteriors share a common property: sensitivity to an example depends on the product of its prediction error and variance (Eq. 12). That is, most sensitive data lies where the model makes the most mistakes and is also least confident. In many cases, such estimates are extremely cheap to compute.

4. We show that sensitivity of the training data can be used to accurately predict model generalization, even during training (Fig. 1(b)). This agrees with similar studies which also show effectiveness of sensitivity in predicting generalization [22, 12, 19, 4].

## 2   Understanding a Model's Sensitivity to Its Training Data

Understanding a model's sensitivity to its training data is important but is often done by a costly process of retraining the model multiple times. For example, consider a model with a parameter vector $\boldsymbol{\theta} \in \mathbb{R}^P$ trained on data $\mathcal{D} = \{\mathcal{D}_1, \mathcal{D}_2, \ldots, \mathcal{D}_N\}$ by using an algorithm $\mathcal{A}_t$ that generates a sequence $\{\boldsymbol{\theta}_t\}$ for iteration $t$ that converges to a minimizer $\boldsymbol{\theta}_*$. Formally, we write

$$\boldsymbol{\theta}_t \leftarrow \mathcal{A}_t\left(\boldsymbol{\theta}_{t-1}, \mathcal{L}(\boldsymbol{\theta})\right) \quad \text{where} \quad \mathcal{L}(\boldsymbol{\theta}) = \sum_{i=1}^{N} \ell_i(\boldsymbol{\theta}) + \mathcal{R}(\boldsymbol{\theta}), \tag{1}$$

and we use the loss $\ell_i(\boldsymbol{\theta})$ for $\mathcal{D}_i$ and a regularizer $\mathcal{R}(\boldsymbol{\theta})$. Because $\boldsymbol{\theta}_t$ are all functions of $\mathcal{D}$ or its subsets, we can analyze their sensitivity by simply 'perturbing' the data. For example, we can remove a subset $\mathcal{M} \subset \mathcal{D}$ to get a perturbed dataset, denoted by $\mathcal{D}^{\backslash \mathcal{M}}$, and retrain the model to get new iterates $\boldsymbol{\theta}_t^{\backslash \mathcal{M}}$, converging to a minimizer $\boldsymbol{\theta}_*^{\backslash \mathcal{M}}$. If the deviation $\boldsymbol{\theta}_t^{\backslash \mathcal{M}} - \boldsymbol{\theta}_t$ is large for most $t$, we may deem the model to be highly sensitive to the examples in $\mathcal{M}$. This is a simple method for sensitive analysis but requires a costly brute-force retraining [38] which is often infeasible for long training trajectories, big models, and large datasets. More importantly, conclusions drawn from retraining are often empirical and may not hold across models or algorithms.

A cheaper alternative is to use local perturbation methods [21], for instance, influence measures that *estimate* the sensitivity without retraining (illustrated in Fig. 1(a) by the dashed red arrow). The simplest result of this kind is for linear regression which dates back to the 70s [7]. The method makes use of the stationarity condition to derive deviations in $\boldsymbol{\theta}_*$ due to small perturbations to data. For linear regression, the deviations can be obtained in closed-form. Consider input-output pairs $(\mathbf{x}_i, y_i)$ and the loss $\ell_i(\boldsymbol{\theta}) = \frac{1}{2}(y_i - f_i(\boldsymbol{\theta}))^2$ for $f_i(\boldsymbol{\theta}) = \mathbf{x}_i^\top \boldsymbol{\theta}$ and a regularizer $\mathcal{R}(\boldsymbol{\theta}) = \delta\|\boldsymbol{\theta}\|^2/2$. We can obtain closed-form expressions of the deviation due to the removal of the $i$'th example as shown below (a proof is included in App. A),

$$\boldsymbol{\theta}_*^{\backslash i} - \boldsymbol{\theta}_* = (\mathbf{H}_*^{\backslash i})^{-1}\mathbf{x}_i e_i, \qquad f_i(\boldsymbol{\theta}_*^{\backslash i}) - f_i(\boldsymbol{\theta}_*) = v_i^{\backslash i} e_i, \tag{2}$$

where we denote $\mathbf{H}_*^{\backslash i} = \mathbf{H}_* - \mathbf{x}_i\mathbf{x}_i^\top$ defined using the Hessian $\mathbf{H}_* = \nabla^2\mathcal{L}(\boldsymbol{\theta}_*)$. We also denote the prediction error of $\boldsymbol{\theta}_*$ by $e_i = \mathbf{x}_i^\top\boldsymbol{\theta}_* - y_i$, and prediction variance of $\boldsymbol{\theta}_*^{\backslash i}$ by $v_i^{\backslash i} = \mathbf{x}_i^\top(\mathbf{H}_*^{\backslash i})^{-1}\mathbf{x}_i$.

The expression shows that the influence is bi-linearly related to both prediction error and variance, that is, when examples with high error and variance are removed, the model is expected to change a lot. These ideas are generalized using *infinitesimal perturbation* [21]. For example, influence functions [8, 32, 31] use a perturbation model $\boldsymbol{\theta}_*^{\epsilon_i} = \arg\min_{\boldsymbol{\theta}} \mathcal{L}(\boldsymbol{\theta}) - \epsilon_i\ell_i(\boldsymbol{\theta})$ with a scalar perturbation $\epsilon_i \in \mathbb{R}$. By using a quadratic approximation, we get the following influence function,

$$\left.\frac{\partial\boldsymbol{\theta}_*^{\epsilon_i}}{\partial\epsilon_i}\right|_{\epsilon_i=0} = \mathbf{H}_*^{-1}\nabla\ell_i(\boldsymbol{\theta}_*). \tag{3}$$

This works for a generic differentiable loss function and is closely related to Eq. 2. We can choose other perturbation models, but they often exhibit bi-linear relationships; see App. A for details.

Despite their generality, there remain many open challenges with the local perturbation methods:

1. Influence functions are valid only at a stationary point $\boldsymbol{\theta}_*$ where the gradient is assumed to be 0, and extending them to iterates $\boldsymbol{\theta}_t$ generated by generic algorithmic-steps $\mathcal{A}_t$ is non-trivial [14]. This is even more important for deep learning where we may never reach such a stationary point, for example, due to stochastic training or early stopping [33, 53].

2. Applying influence functions to a non-differentiable loss or discrete parameter spaces is difficult. This is because the choice of perturbation model is not always obvious [32].

3. Finally, despite their generality, these measures do not directly reveal the causes of high influence. Does the bi-linear relationship in Eq. 2 hold more generally? If yes, under what conditions? Answers to such questions are currently unknown.

Studies to fix these issues are rare in ML, rather it is more common to simply use heuristics measures. Many such measures have been proposed in the recent years, for example, those using derivatives with respect to inputs [23, 2, 38], variations of Cook's distance [17], prediction error and/or gradients [3, 51, 42, 40], backtracking training trajectories [16], or simply by retraining [13]. These works, although useful, do not directly address the issues. Many of these measures are derived without any direct connections to perturbation methods. They also appear to be unaware of bi-linear relationships such as those in Eq. 2. Our goal here is to address the issues by unifying and generalizing perturbation methods of sensitivity analysis.

## 3   The Memory-Perturbation Equation (MPE)

We propose the memory-perturbation equation (MPE) to unify, generalize, and understand sensitivity methods in machine learning. We derive the equation by using a property of *conjugate Bayesian models* which enables us to derive a closed-form expression for the sensitivity. In a Bayesian setting, data examples can be removed by simply dividing their likelihoods from the posterior [52]. For example, consider a model with prior $p_0 = p(\boldsymbol{\theta})$ and likelihood $\tilde{p}_j = p(\mathcal{D}_j|\boldsymbol{\theta})$, giving rise to a posterior $q_* = p(\boldsymbol{\theta}|\mathcal{D}) \propto p_0\tilde{p}_1\tilde{p}_2 \ldots \tilde{p}_N$. To remove $\tilde{p}_j$, say for all $j \in \mathcal{M} \subset \mathcal{D}$, we simply divide $q_*$ by those $\tilde{p}_j$. This is further simplified if we assume conjugate exponential-family form for $p_0$ and $\tilde{p}_j$. Then, the division between two distributions is equivalent to a subtraction between their natural parameters. This property yields a closed-form expression for the exact deviation, as stated below.

**Theorem 1** *Assuming a conjugate exponential-family model, the posterior $q_*^{\backslash\mathcal{M}}$ (with natural parameter $\boldsymbol{\lambda}_*^{\backslash\mathcal{M}}$) can be written in terms of $q_*$ (with natural parameter $\boldsymbol{\lambda}_*$), as shown below:*

$$q_*^{\backslash\mathcal{M}} \propto \frac{q_*}{\prod_{j\in\mathcal{M}} \tilde{p}_j} \quad \implies \quad e^{\langle\boldsymbol{\lambda}_*^{\backslash\mathcal{M}}, \mathbf{T}(\boldsymbol{\theta})\rangle} \propto \frac{e^{\langle\boldsymbol{\lambda}_*, \mathbf{T}(\boldsymbol{\theta})\rangle}}{\prod_{j\in\mathcal{M}} e^{\langle\widetilde{\boldsymbol{\lambda}}_j, \mathbf{T}(\boldsymbol{\theta})\rangle}} \quad \implies \quad \boldsymbol{\lambda}_*^{\backslash\mathcal{M}} = \boldsymbol{\lambda}_* - \sum_{j\in\mathcal{M}} \widetilde{\boldsymbol{\lambda}}_j. \tag{4}$$

*where all exponential families are defined by using inner-product $\langle\boldsymbol{\lambda}, \mathbf{T}(\boldsymbol{\theta})\rangle$ with natural parameters $\boldsymbol{\lambda}$ and sufficient statistics $\mathbf{T}(\boldsymbol{\theta})$. The natural parameter of $\tilde{p}_j$ is denoted by $\widetilde{\boldsymbol{\lambda}}_j$.*

The deviation $\boldsymbol{\lambda}_*^{\backslash\mathcal{M}} - \boldsymbol{\lambda}_*$ is obtained by simply adding $\widetilde{\boldsymbol{\lambda}}_j$ for all $j \in \mathcal{M}$. Further explanations and examples are given in App. B, along with some elementary facts about exponential families. We use this result to derive an equation that enables us to estimate the sensitivity of generic algorithms.

Our derivation builds on the Bayesian learning rule (BLR) [28] which unifies many algorithms by expressing their iterations as inference in conjugate Bayesian models [26]. This is done by reformulating Eq. 1 in a Bayesian setting to find an exponential-family approximation $q_* \approx p(\boldsymbol{\theta}|\mathcal{D}) \propto e^{-\mathcal{L}(\boldsymbol{\theta})}$. At every iteration $t$, the BLR updates the natural parameter $\boldsymbol{\lambda}_t$ of an exponential-family $q_t$ which can equivalently be expressed as the posterior of a conjugate model (shown on the right),

$$\boldsymbol{\lambda}_t \leftarrow (1-\rho)\boldsymbol{\lambda}_{t-1} - \rho\sum_{j=0}^{N}\tilde{\mathbf{g}}_j(\boldsymbol{\lambda}_{t-1}) \quad \Longleftrightarrow \quad q_t \propto \underbrace{(q_{t-1})^{1-\rho}(p_0)^\rho}_{\text{Prior}}\prod_{j=1}^{N}\underbrace{e^{\langle-\rho\tilde{\mathbf{g}}_j(\boldsymbol{\lambda}_{t-1}), \mathbf{T}(\boldsymbol{\theta})\rangle}}_{\text{Likelihood}} \tag{5}$$

where $\tilde{\mathbf{g}}_j(\boldsymbol{\lambda}) = \mathbf{F}(\boldsymbol{\lambda})^{-1}\nabla_{\boldsymbol{\lambda}}\mathbb{E}_q[\ell_j(\boldsymbol{\theta})]$ is the natural gradient with respect to $\boldsymbol{\lambda}$ defined using the Fisher Information Matrix $\mathbf{F}(\boldsymbol{\lambda}_t)$ of $q_t$, and $\rho > 0$ is the learning rate. For simplicity, we denote $\ell_0(\boldsymbol{\theta}) = \mathcal{R}(\boldsymbol{\theta}) = -\log p_0$, and assume $p_0$ to be conjugate. The conjugate model on the right uses a prior and likelihood both of which, by construction, belong to the same exponential-family as $q_t$. By choosing an appropriate form for $q_t$ and making necessary approximations to $\tilde{\mathbf{g}}_j$, the BLR can recover many popular algorithms as special cases. For instance, using a Gaussian $q_t$, we can recover stochastic gradient descent (SGD), Newton's method, RMSprop, Adam, etc. For such cases, the conjugate model at the right is often a linear model [25]. These details, along with a summary of the BLR, are included in App. C. Our main idea is to study the sensitivity of all the algorithms covered under the BLR by using the conjugate model in Eq. 5.

Let $q_t^{\backslash\mathcal{M}}$ be the posterior obtained with the BLR but without the data in $\mathcal{M}$. We can estimate its natural parameter $\boldsymbol{\lambda}_t^{\backslash\mathcal{M}}$ in a similar fashion as Eq. 4, that is, by dividing $q_t$ by the likelihood approximation at the current $\boldsymbol{\lambda}_t$. This gives us the following estimate of the deviation obtained by simply adding the natural-gradients for all examples in $\mathcal{M}$,

$$\hat{\boldsymbol{\lambda}}_t^{\backslash\mathcal{M}} - \boldsymbol{\lambda}_t = \rho\sum_{j\in\mathcal{M}}\tilde{\mathbf{g}}_j(\boldsymbol{\lambda}_t) \tag{6}$$

where $\hat{\boldsymbol{\lambda}}_t^{\backslash\mathcal{M}}$ is an estimate of the true $\boldsymbol{\lambda}_t^{\backslash\mathcal{M}}$. We call this the memory-perturbation equation (MPE) due to a unique property of the equation: the deviation is estimated by a simple addition and characterized solely by the examples in $\mathcal{M}$. Due to the additive nature of the estimate, examples with larger natural-gradients contribute more to it and so we expect most of the sensitivity to be explained by just a few examples with largest natural gradients. This is similar to the representer theorem where just a few support vectors are sufficient to characterize the decision boundary [29, 47, 10]. Here, such examples can be seen as characterizing the model's memory because perturbing them can make the model forget its essential knowledge. The phrase memory-perturbation signifies this.

The equation can be easily adopted to handle an arbitrary perturbation. For instance, consider perturbation $\mathcal{L}(\boldsymbol{\theta}) - \sum_{j\in\mathcal{M}}\epsilon_j\ell_j(\boldsymbol{\theta})$. To estimate its effect, we divide $q_t$ by the likelihood approximations raised to $\epsilon_i$, giving us the following variant,

$$\hat{\boldsymbol{\lambda}}_t^{\boldsymbol{\epsilon}_\mathcal{M}} - \boldsymbol{\lambda}_t = \rho\sum_{j\in\mathcal{M}}\epsilon_j\tilde{\mathbf{g}}_j(\boldsymbol{\lambda}_t), \quad \implies \quad \left.\frac{\partial\hat{\boldsymbol{\lambda}}_t^{\boldsymbol{\epsilon}_\mathcal{M}}}{\partial\epsilon_j}\right|_{\epsilon_j=0} = \rho\tilde{\mathbf{g}}_j(\boldsymbol{\lambda}_t), \quad \forall j \in \mathcal{M}, \tag{7}$$

where we denote all $\epsilon_j$ in $\mathcal{M}$ by $\boldsymbol{\epsilon}_\mathcal{M}$. Setting $\epsilon_j = 1$ in the left reduces to Eq. 6 which corresponds to removal. The example demonstrates how to adopt the MPE to handle arbitrary perturbations.

## 3.1 Unifying the existing sensitivity measures as special cases of the MPE

The MPE is a unifying equation from which many existing sensitivity measures can be derived as special cases. We will show three such results. The first result shows that, for conjugate models, the MPE recovers the exact deviations given in Thm. 1. Such models include textbook examples [6], such as, mixture models, linear state-space models, and PCA. Below is a formal statement.

**Theorem 2** *For conjugate exponential-family models, Eq. 4 is obtained as a special case of the MPE in Eq. 6 evaluated at $\boldsymbol{\lambda}_*$ of the exact posterior $q_*$ when we set $\ell_j(\boldsymbol{\theta}) = -\log \tilde{p}_j$ and $\rho = 1$.*

The result holds because, for conjugate models, one-step of the BLR is equivalent to Bayes' rule and therefore $\tilde{\mathbf{g}}_j(\boldsymbol{\lambda}_*) = -\tilde{\boldsymbol{\lambda}}_j$ (see [24, Sec. 5.1]). A proof is given in App. D along with an illustrative example on the Beta-Bernoulli model. We note that a recent work in [49] also takes inspiration from Bayesian models, but their sensitivity measures lack the property discussed above. See also [15] for a different approach to sensitivity analysis of variational Bayes with a focus on the posterior mean. The result above also justifies setting $\rho$ to 1, a choice we will often resort to.

Our second result is to show that the MPE recovers the influence function by Cook [7].

**Theorem 3** *For linear regression, Eq. 2 is obtained as a special case of the MPE in Eq. 6 evaluated at $\boldsymbol{\lambda}_*$ of the exact posterior $q_* = \mathcal{N}(\boldsymbol{\theta}|\boldsymbol{\theta}_*, \mathbf{H}_*^{-1})$.*

The proof in App. E relies on two facts: first, the natural parameter is $\boldsymbol{\lambda}_* = (\mathbf{H}_*\boldsymbol{\theta}_*, -\frac{1}{2}\mathbf{H}_*)$, and second, the natural gradients for a Gaussian $q$ with mean $\mathbf{m}$ can be written as follows,

$$\tilde{\mathbf{g}}_i(\boldsymbol{\lambda}) = (\hat{\mathbf{g}}_i - \hat{\mathbf{H}}_i\mathbf{m}, \ \tfrac{1}{2}\hat{\mathbf{H}}_i), \tag{8}$$

where $\hat{\mathbf{g}}_i = \mathbb{E}_q[\nabla\ell_i(\boldsymbol{\theta})]$ and $\hat{\mathbf{H}}_i = \mathbb{E}_q[\nabla^2\ell_i(\boldsymbol{\theta})]$. This is due to [28, Eqs. 10-11], but a proof is given in Eq. 27 of App. C. The theorem then directly follows by plugging $\tilde{\mathbf{g}}_i(\boldsymbol{\lambda}_*)$ in Eq. 6. This derivation is much shorter than the classical techniques which often require inversion lemmas (see App. A.1). The estimated deviations are exact, which is not a surprise because linear regression is a conjugate Gaussian model. However, it is interesting (and satisfying) that the deviation in $\boldsymbol{\theta}_*$ naturally emerges from the deviation in $\boldsymbol{\lambda}_*$.

Our final result is to recover influence functions for deep learning, specifically Eq. 3. To do so, we use a Gaussian posterior approximation $q_* = \mathcal{N}(\boldsymbol{\theta}|\boldsymbol{\theta}_*, \mathbf{H}_*^{-1})$ obtained by using the so-called Laplace's method [34, 50, 37]. The Laplace posterior can be seen a special case of the BLR solution when the natural gradient is approximated with the delta method [28, Table 1]. Remarkably, using the same approximation in the MPE, we recover Eq. 3.

**Theorem 4** *The influence function in Eq. 3 is obtained as a special case of the MPE in Eq. 7 evaluated at $\boldsymbol{\lambda}_*$ of the posterior $q_* = \mathcal{N}(\boldsymbol{\theta}|\boldsymbol{\theta}_*, \mathbf{H}_*^{-1})$ when we approximate $\tilde{\mathbf{g}}_i(\boldsymbol{\lambda})$ of Eq. 8 with the delta method by substituting $\mathbb{E}_{q_*}[\nabla\ell_i(\boldsymbol{\theta})] \approx \nabla\ell_i(\boldsymbol{\theta}_*)$ and $\mathbb{E}_{q_*}[\nabla^2\ell_i(\boldsymbol{\theta})] \approx \nabla^2\ell_i(\boldsymbol{\theta}_*)$.*

A proof is in App. F. We note that Eq. 3 can be justified as a Newton-step over the perturbed data but in the opposite direction [32, 31]. In a similar fashion, Eqs. 6 and 7 can be seen as *natural-gradient* steps in the opposite direction. Using the natural-gradient descent, as we have shown, can recover a variety of existing perturbation methods as special cases.

## 3.2 Generalizing the perturbation method to estimate sensitivity during training

Influence measures discussed so far assume that the model is already trained and that the loss is differentiable. We will now present generalizations to obtain *new measures that can be applied during training and do not require differentiability of the loss*. We will focus on Gaussian $q$ but the derivation can be extended to other posterior forms. The main idea is to specialize Eqs. 6 and 7 to the algorithms covered under the BLR, giving rise to new measures that estimate sensitivity by simply taking a step over the perturbed data but in the opposite direction.

We first discuss sensitivity of an iteration $t$ of the BLR yielding a Gaussian $q_t = \mathcal{N}(\boldsymbol{\theta}|\mathbf{m}_t, \mathbf{S}_t^{-1})$. The natural parameter is the pair $\boldsymbol{\lambda}_t = (\mathbf{S}_t\mathbf{m}_t, -\frac{1}{2}\mathbf{S}_t)$. Using Eq. 8 in Eq. 6, we get

$$\hat{\mathbf{S}}_t^{\backslash\mathcal{M}}\hat{\mathbf{m}}_t^{\backslash\mathcal{M}} - \mathbf{S}_t\mathbf{m}_t = \rho \sum_{j\in\mathcal{M}} \mathbb{E}_{q_t}[\nabla\ell_j(\boldsymbol{\theta})] - \mathbb{E}_{q_t}[\nabla^2\ell_j(\boldsymbol{\theta})]\mathbf{m}_t, \quad \mathbf{S}_t - \hat{\mathbf{S}}_t^{\backslash\mathcal{M}} = \rho \sum_{j\in\mathcal{M}} \mathbb{E}_{q_t}[\nabla^2\ell_j(\boldsymbol{\theta})]$$

$$\tag{9}$$

| Algorithm | Update | Sensitivity |
|---|---|---|
| Newton's method | $\boldsymbol{\theta}_t \leftarrow \boldsymbol{\theta}_{t-1} - \mathbf{H}_{t-1}^{-1}\nabla\mathcal{L}(\boldsymbol{\theta}_{t-1})$ | $\mathbf{H}_{t-1}^{-1}\nabla\ell_i(\boldsymbol{\theta}_t)$ |
| Online Newton (ON) [28] | $\boldsymbol{\theta}_t \leftarrow \boldsymbol{\theta}_{t-1} - \rho\,\mathbf{S}_t^{-1}\nabla\mathcal{L}(\boldsymbol{\theta}_{t-1})$ | $\mathbf{S}_t^{-1}\nabla\ell_i(\boldsymbol{\theta}_t)$ |
| ON (diagonal+minibatch) [28] | $\boldsymbol{\theta}_t \leftarrow \boldsymbol{\theta}_{t-1} - \rho\,\mathbf{s}_t^{-1}\cdot\hat{\nabla}\mathcal{L}(\boldsymbol{\theta}_{t-1})$ | $\mathbf{s}_t^{-1}\cdot\nabla\ell_i(\boldsymbol{\theta}_t)$ |
| iBLR (diagonal+minibatch) [35] | $\mathbf{m}_t \leftarrow \mathbf{m}_{t-1} - \rho\,\mathbf{s}_t^{-1}\cdot\hat{\nabla}\mathcal{L}(\boldsymbol{\theta}_{t-1})$ | $\mathbf{s}_t^{-1}\cdot\nabla\ell_i(\boldsymbol{\theta}_t)$ |
| RMSprop/Adam [30] | $\boldsymbol{\theta}_t \leftarrow \boldsymbol{\theta}_{t-1} - \rho\,\mathbf{s}_t^{-\frac{1}{2}}\cdot\hat{\nabla}\mathcal{L}(\boldsymbol{\theta}_{t-1})$ | $\mathbf{s}_t^{-\frac{1}{2}}\cdot\nabla\ell_i(\boldsymbol{\theta}_t)$ |
| SGD | $\boldsymbol{\theta}_t \leftarrow \boldsymbol{\theta}_{t-1} - \rho\,\hat{\nabla}\mathcal{L}(\boldsymbol{\theta}_{t-1})$ | $\nabla\ell_i(\boldsymbol{\theta}_t)$ |

Table 1: A list of algorithms and their sensitivity measures derived using Eq. 10. The second column gives the update, most of which use pre-conditioners that are either matrices ($\mathbf{H}_t, \mathbf{S}_t$) or a vector ($\mathbf{s}_t$); see the full update equations in Eqs. 31 to 34 in App. C. The third column shows the associated sensitivity measure to perturbation in the $i$'th example which can be interpreted as a step for the $i$ example but in the opposite direction. We denote the element-wise multiplication between vectors by "·" and the minibatch gradients by $\hat{\nabla}$. For iBLR, $\boldsymbol{\theta}_t$ is either $\mathbf{m}_t$ or a sample from $q_t$.

Plugging $\mathbf{S}_t$ from the second equation into the first one, we can recover the following expressions,

$$\hat{\mathbf{m}}_t^{\backslash\mathcal{M}} - \mathbf{m}_t = \rho\big(\hat{\mathbf{S}}_t^{\backslash\mathcal{M}}\big)^{-1}\mathbb{E}_{q_t}\Big[\sum_{j\in\mathcal{M}}\nabla\ell_j(\boldsymbol{\theta})\Big], \qquad \frac{\partial\hat{\mathbf{m}}_t^{\epsilon_i}}{\partial\epsilon_i}\bigg|_{\epsilon_i=0} = \rho\,\mathbf{S}_t^{-1}\mathbb{E}_{q_t}\big[\nabla\ell_i(\boldsymbol{\theta})\big] \qquad (10)$$

For the second equation, we omit the proof but it is similar to App. F, resulting in preconditioning with $\mathbf{S}_t$. For computational ease, we will approximate $\hat{\mathbf{S}}_t^{\backslash\mathcal{M}} \approx \mathbf{S}_t$ even in the first equation. We will also approximate the expectation at a sample $\boldsymbol{\theta}_t \sim q_t$ or simply at the mean $\boldsymbol{\theta}_t = \mathbf{m}_t$. Ultimately, the suggestion is to use $\mathbf{S}_t^{-1}\nabla\ell_i(\boldsymbol{\theta}_t)$ as the sensitivity measure, or variations of it, for example, by using a Monte-Carlo average over multiple samples.

Based on this, a list of algorithms and their corresponding measures is given in Table 1. All of the algorithms can be derived as special instances of the BLR by making specific approximations (see App. C.3). The measures are obtained by applying the exact same approximations to Eq. 10. For example, Newton's method is obtained when $\mathbf{m}_t = \boldsymbol{\theta}_t$, $\mathbf{S}_t = \mathbf{H}_{t-1}$, and expectations are approximated by using the delta method at $\boldsymbol{\theta}_t$ (similarly to Thm. 4). With these, we get

$$\mathbf{S}_t^{-1}\mathbb{E}_{q_t}[\nabla\ell_i(\boldsymbol{\theta})] \approx \mathbf{H}_{t-1}^{-1}\nabla\ell_i(\boldsymbol{\theta}_t), \qquad (11)$$

which is the measure shown in the first row of the table. In a similar fashion, we can derive measures for other algorithms that use a slightly different approximations leading to a different preconditioner. The exact strategy to update the preconditioners is given in Eqs. 31 to 34 of App. C.3. For all, the sensitivity measure is simply an update step for the $i$'th example but in the opposite direction.

Table 1 shows an interplay between the training algorithm and sensitivity measures. For instance, it suggests that the measure $\mathbf{H}_{t-1}^{-1}\nabla\ell_i(\boldsymbol{\theta}_t)$ is justifiable for Newton's method but might be inappropriate otherwise. In general, it is more appropriate to use the algorithm's own preconditioner (if they use one). The quality of preconditioner (and therefore the measure) is tied to the quality of the posterior approximation. For example, RMSprop's preconditioner is not a good estimator of the posterior covariance when minibatch size is large [27, Thm. 1], therefore we should not expect it to work well for large minibatches. In contrast, the ON method [28] explicitly builds a good estimate of $\mathbf{S}_t$ during training and we expect it to give better (and more faithful) sensitivity estimates.

For SGD, our approach suggests using the gradient. This goes well with many existing approaches [40, 42, 51, 3] but also gives a straightforward way to modify them when the training algorithm is changed. For instance, the TracIn approach [42] builds sensitivity estimates during SGD training by tracing $\nabla\ell_j(\boldsymbol{\theta}_t)^\top\nabla\ell_i(\boldsymbol{\theta}_t)$ for many examples $i$ and $j$. When the algorithm is switched, say to the ON method, we simply need to trace $\nabla\ell_j(\boldsymbol{\theta}_t)^\top\mathbf{S}_t^{-1}\nabla\ell_i(\boldsymbol{\theta}_t)$. Such a modification is speculated in [42, Sec 3.2] and the MPE provides a way to accomplish exactly that. It is also possible to mix and match algorithms with different measures but caution is required. For example, to use the measure in Eq. 11, say within a first-order method, the algorithm must be modified to build a well-conditioned estimate of the Hessian. This can be tricky and can make the sensitivity measure fragile [5].

Extensions to non-differentiable loss functions and discontinuous parameter spaces is straightforward. For example, when using a Gaussian posterior, the measures in Eq. 10 can be modified to

handle non-differentiable loss function by simply replacing $\mathbb{E}_{q_t}[\nabla \ell_i(\boldsymbol{\theta})]$ with $\nabla_{\mathbf{m}} \mathbb{E}_{q_t}[\ell_i(\boldsymbol{\theta})]$, which is a simple application of the Bonnet theorem [44] (see App. G). The resulting approach is more principled than [32] which uses an ad-hoc smoothing of the non-differentiable loss: the smoothing in our approach is automatically done by using the posterior distribution. Handling of discontinuous parameter spaces follows in a similar fashion. For example, binary variables can be handled by measuring the sensitivity through the parameter of the Bernoulli distribution (see App. D).

### 3.3 Understanding the causes of high sensitivity estimates for the Gaussian case

The MPE can be used to understand the causes of high sensitivity-estimates. We will demonstrate this for Gaussian $q$ but similar analysis can be done for other distributions. We find that sensitivity measures derived using Gaussian posteriors generally have two causes of high sensitivity.

To see this, consider a loss $\ell_i(\boldsymbol{\theta}) = -\log p(y_i|\sigma(f_i(\boldsymbol{\theta})))$ where $p(y_i|\mu)$ is an exponential-family distribution with expectation parameter $\mu$, $f_i(\boldsymbol{\theta})$ is the model output for the $i$'th example, and $\sigma(\cdot)$ is an activation function, for example, the softmax function. For such loss functions, the gradient takes a simple form: $\nabla \ell_i(\boldsymbol{\theta}) = \nabla f_i(\boldsymbol{\theta})[\sigma(f_i(\boldsymbol{\theta})) - y_i]$ [6, Eq. 4.124]. Using this, we can approximate the deviations in model outputs by using a first-order Taylor approximation,

$$\underbrace{f_i(\boldsymbol{\theta}_t^{\backslash i}) - f_i(\boldsymbol{\theta}_t)}_{\text{Deviation in the output}} \approx \nabla f_i(\boldsymbol{\theta}_t)^\top (\boldsymbol{\theta}_t^{\backslash i} - \boldsymbol{\theta}_t) \approx \underbrace{\nabla f_i(\boldsymbol{\theta}_t)^\top \mathbf{H}_{t-1}^{-1} \nabla f_i(\boldsymbol{\theta}_t)}_{=v_{it},\text{ prediction variance}} \underbrace{[\sigma(f_i(\boldsymbol{\theta}_t)) - y_i]}_{=e_{it},\text{ prediction error}}. \tag{12}$$

where we used $\boldsymbol{\theta}_t^{\backslash i} - \boldsymbol{\theta}_t \approx \mathbf{H}_{t-1}^{-1} \nabla \ell_i(\boldsymbol{\theta}_t)$ which is based on the measure in the first row of Table 1. Similarly to Eq. 2, the deviation in the model output is equal to the product of the prediction error and (linearized) prediction variance of $f_i(\boldsymbol{\theta}_t)$ [25, 20]. The change in the model output is expected to be high, whenever examples with high prediction error and variance are removed.

We can write many such variants with a similar bi-linear relationship. For example, Eq. 12 can be extended to get deviations in predictions as follows:

$$\sigma(f_i(\boldsymbol{\theta}_t^{\backslash i})) - \sigma(f_i(\boldsymbol{\theta}_t)) \approx \sigma'(f_i(\boldsymbol{\theta}_t)) \nabla f_i(\boldsymbol{\theta}_t)^\top (\boldsymbol{\theta}_t^{\backslash i} - \boldsymbol{\theta}_t) \approx \sigma'(f_i(\boldsymbol{\theta}_t)) v_{it} e_{it}. \tag{13}$$

Eq. 12 estimates the deviation at one example and at a location $\boldsymbol{\theta}_t$, but we could also write them for a *group* of examples and evaluate them at the mean $\mathbf{m}_t$ or at any sample $\boldsymbol{\theta} \sim q_t$. For example, to remove a group $\mathcal{M}$ of size $M$, we can write the deviation of the model-output vector $\mathbf{f}(\boldsymbol{\theta}) \in \mathbb{R}^M$,

$$\mathbf{f}(\mathbf{m}_t^{\backslash \mathcal{M}}) - \mathbf{f}(\mathbf{m}_t) \approx \nabla \mathbf{f}(\mathbf{m}_t)^\top \mathbf{S}_t^{-1} \nabla \mathbf{f}(\mathbf{m}_t)[\sigma(\mathbf{f}(\mathbf{m}_t) - \mathbf{y}], \tag{14}$$

where $\mathbf{y}$ is the vector of labels and we used the sensitivity measure in Eq. 10. An example for sparse Gaussian process is in App. H. The measure for SGD in Table 1 can also be used which gives $f_i(\boldsymbol{\theta}_t^{\backslash i}) - f_i(\boldsymbol{\theta}_t) \approx \|\nabla f_i(\boldsymbol{\theta})\|^2 e_{it}$ which is similar to the scores used in [40]. The list in Table 1 suggests that such scores can be improved by using $\mathbf{H}_t$ or $\mathbf{S}_t$, essentially, replacing the gradient norm by an estimate of the prediction variance. Additional benefit can be obtained by further employing samples from $q_t$ instead of using a point estimate $\boldsymbol{\theta}_t$ or $\mathbf{m}_t$; see an example in App. H.

It is also clear that all of the deviations above can be obtained cheaply during training by using already computed quantities. The estimation does not add significant computational overhead and can be used to efficiently predict the generalization performance during training. For example, using Eq. 12, we can approximate the leave-one-out (LOO) cross-validation (CV) error as follows,

$$\text{LOO}(\boldsymbol{\theta}_t) = \sum_{i=1}^N \ell_i(\boldsymbol{\theta}_t^{\backslash i}) = -\sum_{i=1}^N \log p(y_i|\sigma(f_i(\boldsymbol{\theta}_t^{\backslash i}))) \approx -\sum_{i=1}^N \log p(y_i|\sigma(f_i(\boldsymbol{\theta}_t) + v_{it} e_{it})). \tag{15}$$

The approximation eliminates the need to train $N$ models to perform CV, rather just uses $e_{it}$ and $v_{it}$ which are extremely cheap to compute within algorithms such as ON, RMSprop, and SGD. Leave-group-out (LGO) estimates can also be built, for example, by using Eq. 14, which enables us to understand the effect of leaving out a big chunk of training data, for example, an entire class for classification. The LOO and LGO estimates are closely related to marginal likelihood and sharpness, both of which are useful to predict generalization performance [22, 12, 19]. Estimates similar to Eq. 15 have been proposed previously [43, 4] but none of them do so during training.

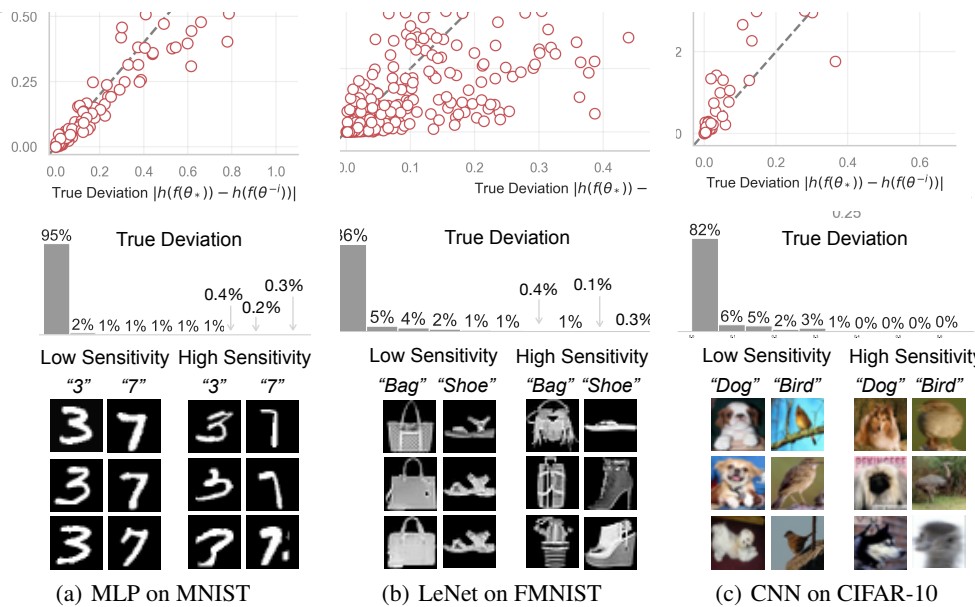

(a) MLP on MNIST      (b) LeNet on FMNIST      (c) CNN on CIFAR-10

Figure 2: The estimated deviation for an example removal correlates well with the true deviations in predictions. Each marker represents an example. For each panel, the histogram at the bottom shows that the majority of examples have low sensitivity and most of the large sensitivities are attributed to a small fraction of data. We show a few images of high and low sensitivity examples from two randomly chosen classes, where we observe the high-sensitivity examples to be more interesting (possibly mislabeled or just ambiguous), while low-sensitivity examples appear more predictable.

## 4 Experiments

We show experimental results to demonstrate the usefulness of the MPE to understand the sensitivity of deep-learning models. We show the following: (1) we verify that the estimated deviations (sensitivities) for data removal correlate with the truth; (2) we predict the effect of class removal on generalization error; (3) we estimate the cross-validation curve for hyperparameter tuning; (4) we predict generalization during training; and (5) we study evolution of sensitivities during training. All details of the experimental setup are included in App. I and the code is available at `https://github.com/team-approx-bayes/memory-perturbation`.

**Estimated deviations correlate with the truth:** Fig. 2 shows a good correlation between the true deviations $\sigma(f_i(\boldsymbol{\theta}_*^{\backslash i})) - \sigma(f_i(\boldsymbol{\theta}_*))$ and their estimates $\sigma'(f_i(\boldsymbol{\theta}_*))v_{i*}e_{i*}$, as shown in Eq. 13. We show results for three datasets, each using a different architecture but all trained using SGD. To estimate the Hessian $\mathbf{H}_*$ and compute $v_{i*} = \nabla f_i(\boldsymbol{\theta}_*)^\top \mathbf{H}_*^{-1} \nabla f_i(\boldsymbol{\theta}_*)$, we use a Kronecker-factored (K-FAC) approximation implemented in the `laplace` [11] and `ASDL` [39] packages. Each marker represents a data example. The estimate roughly maintains the ranking of examples according to their sensitivity. Below each panel, a histogram of true deviations is included to show that the majority of examples have extremely low sensitivity and most of the large sensitivities are attributed to a small fraction of data. The high-sensitivity examples often include interesting cases (possibly mislabeled or simply ambiguous), some of which are visualized in each panel along with some low-sensitivity examples to show the contrast. High-sensitivity examples characterize the model's memory because perturbing them leads to a large change in the model. Similar trends are observed for removal of *groups* of examples in Fig. 6 of App. I.2.

**Predicting the effect of class removal on generalization:** Fig. 3(a) shows that the leave-group-out estimates can be used to faithfully predict the test performance even when a *whole class* is removed. The x-axis shows the test negative log-likelihood (NLL) on a held-out test set, while the y-axis shows the following leave-one-class-out (LOCO) loss on the set $\mathcal{C}$ of a left-out class,

$$\text{LOCO}_{\mathcal{C}}(\boldsymbol{\theta}_*) = \sum_{i \in \mathcal{C}} \ell_i(\boldsymbol{\theta}_*^{\backslash \mathcal{C}}) \approx - \sum_{i \in \mathcal{C}} \log p(y_i | \sigma(f_i(\boldsymbol{\theta}_*) + v_{i*}e_{i*})).$$

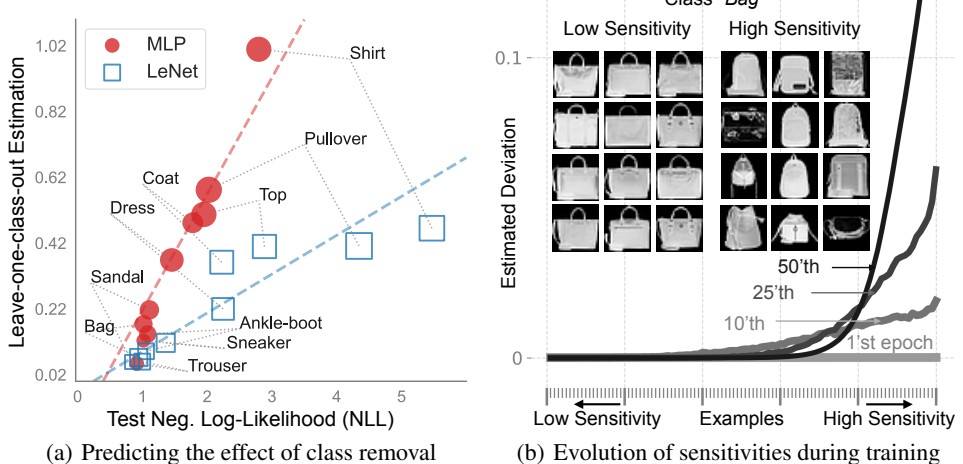

(a) Predicting the effect of class removal

(b) Evolution of sensitivities during training

Figure 3: Panel (a) shows, in the x-axis, the test NLL of trained models with a class removed. In the y-axis, we show the respective leave-one-class-out (LOCO) estimates. Each marker correspond to a specific class removed (text indicates class names). Results for two models on FMNIST are shown. Both show good correlation between the test NLL and LOCO estimates; see the dashed lines. Panel (b) shows the evolution of estimated sensitivities during training of LeNet5 on FMNIST. As training progresses, the model becomes more and more sensitive to a small fraction of data.

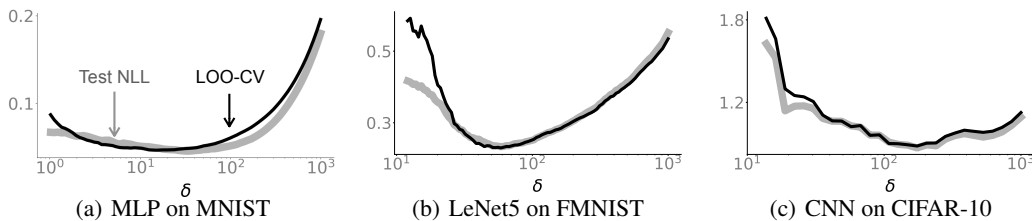

(a) MLP on MNIST  (b) LeNet5 on FMNIST  (c) CNN on CIFAR-10

Figure 4: The test NLL (gray) almost perfectly matches the estimated LOO-CV error of Eq. 15 (black). The x-axis shows different values of $\delta$ parameter of an $L_2$-regularization $\delta\|\boldsymbol{\theta}\|^2/2$.

The estimate uses an approximation: $f_i(\boldsymbol{\theta}_*^{\setminus\mathcal{C}}) - f_i(\boldsymbol{\theta}_*) \approx \nabla f_i(\boldsymbol{\theta}_*)^\top \mathbf{H}_*^{-1} \sum_{j\in\mathcal{C}} \nabla\ell_j(\boldsymbol{\theta}_*) \approx v_{i*}e_{i*}$, which is similar to Eq. 12, but uses an additional approximation $\sum_{j\in\mathcal{C}} \nabla\ell_j(\boldsymbol{\theta}_*) \approx \nabla\ell_i(\boldsymbol{\theta}_*)$ to reduce the computation due to matrix-vector multiplications (we rely on the same K-FAC approximation used in the previous experiment). Results might improve when this approximation is relaxed. We show results for two models: MLP and LeNet. Each marker corresponds to a specific class whose names are indicated with the text. The dashed lines indicate the general trends, showing a good correlation between the truth and estimate. The classes *Shirt*, *Pullover* are the most sensitive, while the classes *Bag*, *Trousers* are least sensitive. A similar result for MNIST is in Fig. 11(d) of App. I.3.

**Predicting generalization for hyperparameter tuning:** We consider the tuning of the parameter $\delta$ for the $L_2$-regularizer of form $\delta\|\boldsymbol{\theta}\|^2/2$. Fig. 4 shows an almost perfect match between the test NLL and the estimated LOO-CV error of Eq. 15. Additional figures with the test errors visualized on top are included in Fig. 7 of App. I.4 where we again see a close match to the LOO-CV curves.

**Predicting generalization during training:** As discussed earlier, existing influence measures are not designed to analyze sensitivity during training and care needs to be taken when using ad-hoc strategies. We first show results for our proposed measure in Eq. 10 which gives reliable sensitivity estimates during training. We use the improved-BLR method [35] which estimates the mean $\mathbf{m}_t$ and a vector preconditioner $\mathbf{s}_t$ during training. We can derive an estimate for the LOO error at the mean $\mathbf{m}_t$ following a derivation similar to Eqs. 14 and 15,

$$\text{LOO}(\mathbf{m}_t) \approx -\sum_{i=1}^{N} \log p(y_i|\sigma(f_i(\mathbf{m}_t) + v_{it}e_{it})) \tag{16}$$

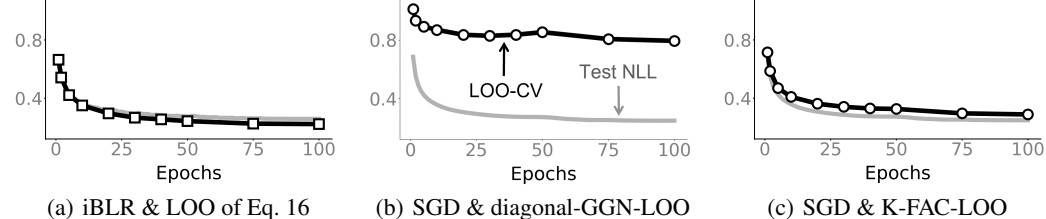

|                            |                                  |                       |
| :------------------------: | :------------------------------: | :-------------------: |
| (a) iBLR & LOO of Eq. 16   | (b) SGD & diagonal-GGN-LOO       | (c) SGD & K-FAC-LOO   |

Figure 5: We compare faithfulness of LOO estimates during training to predict the test NLL. The first panel shows results for iBLR where a good match is obtained by using the LOO estimate of Eq. 16 which uses a diagonal preconditioner. The next two panels show results for SGD where we use the LOO estimate of Eq. 15 but with different Hessian approximations. Panel (b) uses a diagonal-GGN which does not work very well. Results are improved when K-FAC is used, but they are still not as good as the iBLR, despite using a non-diagonal Hessian approximation.

where $v_{it} = \nabla f_i(\mathbf{m}_t)^\top \text{diag}(\mathbf{s}_t)^{-1} \nabla f_i(\mathbf{m}_t)$ and $e_{it} = \sigma(f_i(\mathbf{m}_t)) - y_i$.

The first panel in Fig. 5 shows a good match between the above LOO estimate and test NLL. For comparison, in the next two panels, we show results for SGD training by using two ad-hoc measures obtained by plugging different Hessian approximations in Eq. 11. The first panel approximates $\mathbf{H}_t$ with a diagonal Generalized Gauss-Newton (GGN) matrix, while the second panel uses a K-FAC approximation. We see that diagonal-GGN-LOO does not work well at all and, while K-FAC-LOO improves this, it is still not as good as the iBLR result despite using a non-diagonal Hessian approximation. Not to mention, the two measures require an additional pass through the data to compute the Hessian approximation, and also need a careful setting of a damping parameter.

A similar result for iBLR is shown in Fig. 1(b) where we use the larger ResNet–20 on CIFAR10, and more such results are included in Fig. 8 of App. I.5. We also find that both diagonal-GGN-LOO or K-FAC-LOO further deteriorate when the model overfits; see Fig. 9. Results for the Adam optimizer are included in Fig. 10, where we again see that using ad hoc measures may not always work. Overall, these results show the difficulty of estimating sensitivity during training and suggest to take caution when using measures that are not naturally suited to analyze the training algorithm.

**Evolution of sensitivities during training:** Fig. 3(b) shows the evolution of sensitivities of examples as the training progresses. We use the iBLR algorithm and approximate the deviation as $\sigma(f_i(\mathbf{m}_t^{\backslash i})) - \sigma(f_i(\mathbf{m}_t)) \approx \sigma'(f_i(\mathbf{m}_t))v_{it}e_{it}$ where $v_{it}$ and $e_{it}$ are obtained similarly to Eq. 16. The x-axis corresponds to examples sorted from least sensitive to most sensitive examples at convergence. The y-axis shows the histogram of sensitivity estimates. We observe that, as the training progresses, the distribution concentrates around a small fraction of the data. At the top, we visualize a few examples with high and low sensitivity estimates, where the high-sensitivity examples included interesting cases (similarly to Fig. 2). The result suggests that the model concentrates more and more on a small fraction of high-sensitivity examples, and therefore such examples can be used to characterize the model's memory. Additional experiments of this kind are included in Fig. 11 of App. I.6, along with other experiment details.

## 5   Discussion

We present the memory-perturbation equation by building upon the BLR framework. The equation suggests to take a step in the direction of the natural gradient of the perturbed examples. Using the MPE framework, we unify existing influence measures, generalize them to a wide variety of problems, and unravel useful properties regarding sensitivity. We also show that sensitivity estimation can be done cheaply and use this to predict generalization performance. An interesting avenue for future research is to apply the method to larger models and real-world problems. We also need to understand how our generalization measure compares to other methods, such as those considered in [22]. We would also like to understand the effect of various posterior approximations. Another interesting direction is to apply the method to non-Gaussian cases, for example, to study ensemble methods in deep learning with mixture models.

**Acknowledgements**

This work is supported by the Bayes duality project, JST CREST Grant Number JPMJCR2112.

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

# A  Influence Function for Linear Regression

We consider $N$ input-output pairs $(\mathbf{x}_i, y_i)$. The feature matrix containing $\mathbf{x}_i^\top$ as rows is denoted by $\mathbf{X}$ and the output vector of length $N$ is denoted by $\mathbf{y}$. The loss is $\ell_i(\boldsymbol{\theta}) = \frac{1}{2}(y_i - f_i(\boldsymbol{\theta}))^2$ for $f_i(\boldsymbol{\theta}) = \mathbf{x}_i^\top \boldsymbol{\theta}$. The regularizer is assumed to be $\mathcal{R}(\boldsymbol{\theta}) = \delta\|\boldsymbol{\theta}\|^2/2$. The minimizer is given by

$$\boldsymbol{\theta}_* = \mathbf{H}_*^{-1}\mathbf{X}^\top \mathbf{y}. \tag{17}$$

We define a perturbation model as follows with $\epsilon_i \in \mathbb{R}$:

$$\boldsymbol{\theta}_*^{\epsilon_i} = \arg\min_{\boldsymbol{\theta}} \mathcal{L}(\boldsymbol{\theta}) - \epsilon_i \ell_i(\boldsymbol{\theta}).$$

For $\epsilon_i = 1$, it corresponds to example removal. An arbitrary $\epsilon_i$ simply weights the example accordingly. The solution has a closed-form expression,

$$\boldsymbol{\theta}_*^{\epsilon_i} = \left(\mathbf{H}_* - \epsilon_i \mathbf{x}_i \mathbf{x}_i^\top\right)^{-1}\left(\mathbf{X}^\top \mathbf{y} - \epsilon_i \mathbf{x}_i y_i\right). \tag{18}$$

where $\mathbf{H}_* = \sum_{i=1}^N \mathbf{x}_i \mathbf{x}_i^\top + \delta \mathbf{I}_P$ is the Hessian of $\mathcal{L}(\boldsymbol{\theta})$. We first derive a closed-form expressions for $\boldsymbol{\theta}_*^{\epsilon_i} - \boldsymbol{\theta}_*$, and then specialize them for different $\epsilon_i$.

## A.1  Derivation of the leave-one-out (LOO) deviation

We denote $\boldsymbol{\Sigma}_* = \mathbf{H}_*^{-1}$ and use the Sherman-Morrison formula to write

$$
\begin{aligned}
\boldsymbol{\theta}_*^{\epsilon_i} &= \left(\boldsymbol{\Sigma}_* + \frac{\epsilon_i \boldsymbol{\Sigma}_* \mathbf{x}_i \mathbf{x}_i^\top \boldsymbol{\Sigma}_*}{1 - \epsilon_i \mathbf{x}_i^\top \boldsymbol{\Sigma}_* \mathbf{x}_i}\right)\left(\mathbf{X}^\top \mathbf{y} - \epsilon_i y_i \mathbf{x}_i\right) \\
&= \boldsymbol{\Sigma}_* \mathbf{X}^\top \mathbf{y} + \epsilon_i \boldsymbol{\Sigma}_* \mathbf{x}_i \left[\frac{\mathbf{x}_i^\top \boldsymbol{\Sigma}_* \mathbf{X}^\top \mathbf{y}}{1 - \epsilon_i \mathbf{x}_i^\top \boldsymbol{\Sigma}_* \mathbf{x}_i} - \frac{\epsilon_i y_i \mathbf{x}_i^\top \boldsymbol{\Sigma}_* \mathbf{x}_i}{1 - \epsilon_i \mathbf{x}_i^\top \boldsymbol{\Sigma}_* \mathbf{x}_i} - y_i\right] \\
&= \boldsymbol{\theta}_* + \epsilon_i \boldsymbol{\Sigma}_* \mathbf{x}_i \left[\frac{\mathbf{x}_i^\top \boldsymbol{\theta}_*}{1 - \epsilon_i v_i} - \frac{\epsilon_i y_i v_i}{1 - \epsilon_i v_i} - y_i\right] \\
&= \boldsymbol{\theta}_* + \epsilon_i \boldsymbol{\Sigma}_* \mathbf{x}_i \left[\frac{\mathbf{x}_i^\top \boldsymbol{\theta}_* - y_i}{1 - \epsilon_i v_i}\right] = \boldsymbol{\theta}_* + \boldsymbol{\Sigma}_* \mathbf{x}_i \frac{\epsilon_i e_i}{1 - \epsilon_i v_i}.
\end{aligned} \tag{19}
$$

In line 3 we substitute $v_i = \mathbf{x}_i^\top \boldsymbol{\Sigma}_* \mathbf{x}_i$ and $\boldsymbol{\theta}_* = \boldsymbol{\Sigma}_* \mathbf{X}^\top \mathbf{y}$ and in the last step we use $e_i = \mathbf{x}_i^\top \boldsymbol{\theta}_* - y_i$.

We define $e_i^{\backslash i} = e_i/(1 - v_i)$ which is the prediction error of $\boldsymbol{\theta}_*^{\backslash i}$,

$$e_i^{\backslash i} = \mathbf{x}_i^\top \boldsymbol{\theta}_*^{\backslash i} - y_i = \mathbf{x}_i^\top \left(\boldsymbol{\theta}_* + \boldsymbol{\Sigma}_* \mathbf{x}_i \frac{e_i}{1 - v_i}\right) - y_i = \mathbf{x}_i^\top \boldsymbol{\theta}_* + \frac{v_i}{1 - v_i} e_i - y_i = \frac{e_i}{1 - v_i}. \tag{20}$$

Therefore, we get the following expressions for the deviation,

$$\boldsymbol{\theta}_*^{\backslash i} - \boldsymbol{\theta}_* = \boldsymbol{\Sigma}_* \mathbf{x}_i e_i^{\backslash i}, \qquad f_i(\boldsymbol{\theta}_*^{\backslash i}) - f_i(\boldsymbol{\theta}_*) = v_i e_i^{\backslash i}.$$

These expressions can be written in the form of Eq. 2 by left-multiplying with $\boldsymbol{\Sigma}_*^{-1} = \mathbf{H}_*^{\backslash i} + \mathbf{x}_i \mathbf{x}_i^\top$,

$$(\mathbf{H}_*^{\backslash i} + \mathbf{x}_i \mathbf{x}_i^\top)(\boldsymbol{\theta}_*^{\backslash i} - \boldsymbol{\theta}_*) = \mathbf{x}_i(\mathbf{x}_i^\top \boldsymbol{\theta}_*^{\backslash i} - y_i) \Rightarrow \boldsymbol{\theta}_*^{\backslash i} - \boldsymbol{\theta}_* = (\mathbf{H}_*^{\backslash i})^{-1} \mathbf{x}_i e_i.$$

## A.2  Derivation of the infinitesimal perturbation approach

We differentiate $\boldsymbol{\theta}_*^{\epsilon_i}$ in Eq. 19 to get

$$\frac{\partial \boldsymbol{\theta}_*^{\epsilon_i}}{\partial \epsilon_i} = \boldsymbol{\Sigma}_* \mathbf{x}_i \frac{e_i}{(1 - \epsilon_i v_i)^2}, \tag{21}$$

yielding the following expressions:

$$\left.\frac{\partial \boldsymbol{\theta}_*^{\epsilon_i}}{\partial \epsilon_i}\right|_{\epsilon_i=0} = \boldsymbol{\Sigma}_* \mathbf{x}_i e_i, \qquad \left.\frac{\partial f_i(\boldsymbol{\theta}_*^{\epsilon_i})}{\partial \epsilon_i}\right|_{\epsilon_i=0} = \mathbf{x}_i^\top \left.\frac{\partial \boldsymbol{\theta}_*^{\epsilon_i}}{\partial \epsilon_i}\right|_{\epsilon_i=0} = \mathbf{x}_i^\top \boldsymbol{\Sigma}_* \mathbf{x}_i e_i = v_i e_i. \tag{22}$$

The second equation in Eq. 22 follows from the chain rule. We get a bi-linear relationship of the influence measure with respect to $v_i$ and prediction error $e_i$. It is also possible to evaluate Eq. 21 at $\epsilon_i = 1$ representing an infinitesimal perturbation about the LOO estimate, $\partial \boldsymbol{\theta}_*^{\epsilon_i}/\partial \epsilon_i|_{\epsilon_i=1} = \boldsymbol{\Sigma}_*^{\backslash i} \mathbf{x}_i e_i^{\backslash i}$ From this and Eq. 22, we can interpret Eq. 2 as the average derivative over the interval $\epsilon_i \in [0, 1]$ [9] or the derivative evaluated at some $0 < \epsilon_i < 1$ (via an application of the mean value theorem) [41].

# B Conjugate Exponential-Family Models

Exponential-family distributions take the following form:

$$q = h(\boldsymbol{\theta}) \exp\left[\langle \boldsymbol{\lambda}, \mathbf{T}(\boldsymbol{\theta}) \rangle - A(\boldsymbol{\lambda})\right].$$

where $\boldsymbol{\lambda} \in \Omega$ are the natural (or canonical) parameter for which the cumulant (or log partition) function $A(\boldsymbol{\lambda})$ is finite, strictly convex and differentiable over $\Omega$. The quantity $\mathbf{T}(\boldsymbol{\theta})$ is the sufficient statistics, $\langle \cdot, \cdot \rangle$ is an inner product and $h(\boldsymbol{\theta})$ is some function. A popular example is the Gaussian distribution, which can be rearranged to take an exponential-family form written in terms of the precision matrix $\mathbf{S} = \boldsymbol{\Sigma}^{-1}$,

$$\mathcal{N}(\boldsymbol{\theta}|\mathbf{m}, \boldsymbol{\Sigma}) = |2\pi\boldsymbol{\Sigma}|^{-\frac{1}{2}} \exp\left[-\tfrac{1}{2}(\boldsymbol{\theta}-\mathbf{m})^\top \boldsymbol{\Sigma}^{-1}(\boldsymbol{\theta}-\mathbf{m})\right]$$

$$= \exp\left[\boldsymbol{\theta}^\top \mathbf{Sm} - \tfrac{1}{2}\boldsymbol{\theta}^\top \mathbf{S}\boldsymbol{\theta} - \tfrac{1}{2}\left(\mathbf{m}^\top \mathbf{Sm} + \log|2\pi\mathbf{S}^{-1}|\right)\right].$$

From this, we can read-off the quantities needed to define an exponential-form,

$$\boldsymbol{\lambda} = (\mathbf{Sm}, -\tfrac{1}{2}\mathbf{S}), \quad \mathbf{T}(\boldsymbol{\theta}) = (\boldsymbol{\theta}, \boldsymbol{\theta}\boldsymbol{\theta}^\top), \quad A(\boldsymbol{\lambda}) = \tfrac{1}{2}\left(\mathbf{m}^\top \mathbf{Sm} + \log|2\pi\mathbf{S}^{-1}|\right), \quad h(\boldsymbol{\theta}) = 1. \quad (23)$$

Both the natural parameter and sufficient statistics consist of two elements. The inner-product for the first elements is simply a transpose to get the $\boldsymbol{\theta}^\top \mathbf{Sm}$ term, while for the second element it is a trace which gives $-\mathrm{Tr}(\boldsymbol{\theta}\boldsymbol{\theta}^\top \mathbf{S}/2) = -\tfrac{1}{2}\boldsymbol{\theta}^\top \mathbf{S}\boldsymbol{\theta}$.

Conjugate Exponential-Family Models are those where both the likelihoods and prior can be expressed in terms of the same form of exponential-family distribution with respect to $\boldsymbol{\theta}$. For instance, in linear regression, both the likelihood and prior take a Gaussian form with respect to $\boldsymbol{\theta}$,

$$\tilde{p}_i = p(y_i|\mathbf{x}_i, \boldsymbol{\theta}) = \mathcal{N}(y_i|\mathbf{x}_i^\top \boldsymbol{\theta}, 1) \propto \exp\left[\boldsymbol{\theta}^\top \mathbf{x}_i y_i - \tfrac{1}{2}\boldsymbol{\theta}^\top \mathbf{x}_i \mathbf{x}_i^\top \boldsymbol{\theta}\right]$$

$$p_0 = p(\boldsymbol{\theta}) = \mathcal{N}(\boldsymbol{\theta}|0, \mathbf{I}/\delta) \propto \exp\left[-\tfrac{1}{2}\boldsymbol{\theta}^\top (\delta\mathbf{I}) \boldsymbol{\theta}\right].$$

Note that $\tilde{p}_i$ is a distribution over $y_i$ but it can also be expressed in an (unnormalized) Gaussian form with respect to $\boldsymbol{\theta}$. The sufficient statistics of both $\tilde{p}_i$ and $p_0$ correspond to those of a Gaussian distribution. Therefore, the posterior is also a Gaussian,

$$q_* = p(\boldsymbol{\theta}|\mathcal{D}) \propto p_0 \tilde{p}_1 \tilde{p}_2 \ldots \tilde{p}_N$$

$$= \exp\left[-\tfrac{1}{2}\boldsymbol{\theta}^\top (\delta\mathbf{I}) \boldsymbol{\theta}\right] \prod_{i=1}^N \exp\left[\boldsymbol{\theta}^\top \mathbf{x}_i y_i - \tfrac{1}{2}\boldsymbol{\theta}^\top \mathbf{x}_i \mathbf{x}_i^\top \boldsymbol{\theta}\right]$$

$$= \exp\left[\boldsymbol{\theta}^\top \sum_{i=1}^N \mathbf{x}_i y_i - \tfrac{1}{2}\boldsymbol{\theta}^\top \left(\delta\mathbf{I} + \sum_{i=1}^N \mathbf{x}_i \mathbf{x}_i^\top\right) \boldsymbol{\theta}\right]$$

$$= \exp\left[\boldsymbol{\theta}^\top \mathbf{H}_* \boldsymbol{\theta}_* - \tfrac{1}{2}\boldsymbol{\theta}^\top \mathbf{H}_* \boldsymbol{\theta}\right]$$

$$\propto \mathcal{N}(\boldsymbol{\theta}|\boldsymbol{\theta}_*, \mathbf{H}_*^{-1}).$$

The third line follows because $\boldsymbol{\theta}_* = \mathbf{H}_*^{-1}\mathbf{X}^\top \mathbf{y}$, as shown in Eq. 17.

These computations can be written as *conjugate-computations* [26] where we simply add the natural parameters,

$$\tilde{p}_i \propto \exp\left[\langle \widetilde{\boldsymbol{\lambda}}_i, \mathbf{T}(\boldsymbol{\theta}) \rangle\right], \text{ where } \widetilde{\boldsymbol{\lambda}}_i = (\mathbf{x}_i y_i, -\tfrac{1}{2}\mathbf{x}_i \mathbf{x}_i^\top)$$

$$p_0 \propto \exp\left[\langle \boldsymbol{\lambda}_0, \mathbf{T}(\boldsymbol{\theta}) \rangle\right], \text{ where } \boldsymbol{\lambda}_0 = (0, -\tfrac{1}{2}\delta\mathbf{I})$$

$$\implies q_* \propto \exp\left[\langle \boldsymbol{\lambda}_*, \mathbf{T}(\boldsymbol{\theta}) \rangle\right], \text{ where } \boldsymbol{\lambda}_* = \boldsymbol{\lambda}_0 + \sum_{i=1}^N \widetilde{\boldsymbol{\lambda}}_i = \left(\mathbf{H}_* \boldsymbol{\theta}_*, -\tfrac{1}{2}\mathbf{H}_*\right).$$

In the same fashion, to remove the contributions of certain likelihoods, we can simply subtract their natural parameters from $\boldsymbol{\lambda}_*$. These are the calculations which give rise to the following equation:

$$q_*^{\backslash\mathcal{M}} \propto \frac{q_*}{\prod_{j\in\mathcal{M}} \tilde{p}_j} \quad \implies e^{\langle \mathbf{T}(\boldsymbol{\theta}), \boldsymbol{\lambda}_*^{\backslash\mathcal{M}} \rangle} \propto \frac{e^{\langle \mathbf{T}(\boldsymbol{\theta}), \boldsymbol{\lambda}_* \rangle}}{\prod_{j\in\mathcal{M}} e^{\langle \mathbf{T}(\boldsymbol{\theta}), \widetilde{\boldsymbol{\lambda}}_j \rangle}} \quad \implies \boldsymbol{\lambda}_*^{\backslash\mathcal{M}} = \boldsymbol{\lambda}_* - \sum_{j\in\mathcal{M}} \widetilde{\boldsymbol{\lambda}}_j.$$

## C  The Bayesian Learning Rule

The Bayesian learning rule (BLR) aims to find a posterior approximation $q(\boldsymbol{\theta}) \approx p(\boldsymbol{\theta}|\mathcal{D}) \propto e^{-\mathcal{L}(\boldsymbol{\theta})}$. Often, one considers a regular, minimal exponential-family $q \in \mathcal{Q}$, for example, the class of Gaussian distributions. The approximation is found by optimizing a generalized Bayesian objective,

$$q_* = \underset{q \in \mathcal{Q}}{\arg\min}\ \mathbb{E}_q\left[\mathcal{L}(\boldsymbol{\theta})\right] - \mathcal{H}(q).$$

where $\mathcal{H}(q) = \mathbb{E}_q[-\log q(\boldsymbol{\theta})]$ is the entropy of $q$ and $\mathcal{Q}$ is the class of exponential family approximation. The objective is equivalent to the Evidence Lower Bound (ELBO) when $\mathcal{L}(\boldsymbol{\theta})$ corresponds to the negative log-joint probability of a Bayesian model; see [28, Sec 1.2].

The BLR uses natural-gradient descent to find $q_*$, where each iteration $t$ takes the following form,

$$\boldsymbol{\lambda}_t \leftarrow \boldsymbol{\lambda}_{t-1} - \rho\mathbf{F}(\boldsymbol{\lambda}_{t-1})^{-1}\left.\frac{\partial}{\partial\boldsymbol{\lambda}}\left[\mathbb{E}_q\left[\mathcal{L}(\boldsymbol{\theta})\right] - \mathcal{H}(q)\right]\right|_{\boldsymbol{\lambda}=\boldsymbol{\lambda}_{t-1}} \tag{24}$$

where $\rho > 0$ is the learning rate. The gradient is computed with respect to $\boldsymbol{\lambda}$ (through $q$), and we scale the gradient by the Fisher Information Matrix (FIM) defined as follows,

$$\mathbf{F}(\boldsymbol{\lambda}) = \mathbb{E}_q\left[(\nabla_{\boldsymbol{\lambda}}\log q)(\nabla_{\boldsymbol{\lambda}}\log q)^{\top}\right] = \nabla_{\boldsymbol{\lambda}}^2 A(\boldsymbol{\lambda}).$$

The second equality shows that, for exponential-family distribution, the above FIM is also the second derivative of the log-partition function $A(\boldsymbol{\lambda})$.

### C.1  The BLR of Eq. 5

The BLR in Eq. 5 is obtained by simplifying the natural-gradient using the following identity,

$$\mathbf{F}(\boldsymbol{\lambda})^{-1}\nabla_{\boldsymbol{\lambda}}\mathbb{E}_q(\cdot) = \left.\nabla_{\boldsymbol{\mu}}\mathbb{E}_q(\cdot)\right|_{\boldsymbol{\mu}=\nabla_{\boldsymbol{\lambda}}A(\boldsymbol{\lambda})} \tag{25}$$

where $\boldsymbol{\mu}$ is the expectation parameter. The identity works because of the minimality of the exponential-family which ensures that there is a one-to-one mapping between $\boldsymbol{\lambda}$ and $\boldsymbol{\mu}$, and also that the FIM is invertible. Using this, we can show that the natural gradient of $\mathcal{H}(q)$ is simply equal to $-\boldsymbol{\lambda}$; see [28, App. B]. Defining $\ell_0(\boldsymbol{\theta}) = \mathcal{R}(\boldsymbol{\theta})$, we get the version of the BLR shown in Eq. 5,

$$\boldsymbol{\lambda}_t \leftarrow (1-\rho)\boldsymbol{\lambda}_{t-1} - \rho\sum_{j=0}^N \tilde{\mathbf{g}}_j(\boldsymbol{\lambda}_{t-1}), \text{ where } \tilde{\mathbf{g}}_j(\boldsymbol{\lambda}_{t-1}) = \left.\nabla_{\boldsymbol{\mu}}\mathbb{E}_q[\ell_j(\boldsymbol{\theta})]\right|_{\boldsymbol{\mu}=\nabla_{\boldsymbol{\lambda}}A(\boldsymbol{\lambda}_{t-1})}.$$

### C.2  The conjugate-model form of the BLR given in Eq. 5

To express the update in terms of the posterior of a conjugate model, we simply take the inner product with $\mathbf{T}(\boldsymbol{\theta})$ and take the exponential to write the update as

$$\underbrace{e^{\langle\boldsymbol{\lambda}_t,\mathbf{T}(\boldsymbol{\theta})\rangle}}_{\propto q_t} \leftarrow \left(\underbrace{e^{\langle\boldsymbol{\lambda}_{t-1},\mathbf{T}(\boldsymbol{\theta})\rangle}}_{\propto q_{t-1}}\right)^{1-\rho}\left(\underbrace{e^{\langle-\tilde{\mathbf{g}}_0(\boldsymbol{\lambda}_{t-1}),\mathbf{T}(\boldsymbol{\theta})\rangle}}_{\propto p_0}\right)^{\rho}\prod_{j=1}^N e^{\langle-\rho\tilde{\mathbf{g}}_j(\boldsymbol{\lambda}_{t-1}),\mathbf{T}(\boldsymbol{\theta})\rangle}, \tag{26}$$

The simplification of the second term on the left to $p_0$ happens when $p_0$ is a conjugate prior, that is, $p_0 \propto \exp(\langle\boldsymbol{\lambda}_0,\mathbf{T}(\boldsymbol{\theta})\rangle)$ for some $\boldsymbol{\lambda}_0$ (see an example in App. B where we show that $L_2$ regularizer leads to such a choice). In such cases, we can simplify,

$$\langle-\tilde{\mathbf{g}}_0(\boldsymbol{\lambda}),\mathbf{T}(\boldsymbol{\theta})\rangle = \langle\nabla_{\boldsymbol{\mu}}\mathbb{E}_q[\log p_0],\mathbf{T}(\boldsymbol{\theta})\rangle = \langle\nabla_{\boldsymbol{\mu}}\langle\boldsymbol{\lambda}_0,\boldsymbol{\mu}\rangle,\mathbf{T}(\boldsymbol{\theta})\rangle = \langle\boldsymbol{\lambda}_0,\mathbf{T}(\boldsymbol{\theta})\rangle = \log p_0 + \text{const.}$$

Using this in Eq. 26, we recover the conjugate model given in Eq. 5.

### C.3  BLR for a Gaussian $q$ and the Variational Online Newton (VON) algorithm

By choosing an appropriate form for $q_t$ and making necessary approximations to $\tilde{\mathbf{g}}_j$, the BLR can recover many popular algorithms as special cases. We will now give a few examples for the case of a Gaussian $q_t = \mathcal{N}(\boldsymbol{\theta}|\mathbf{m}_t,\boldsymbol{\Sigma}_t)$ which enables derivation of various first and second-order optimization algorithms, such as, Newton's method, RMSprop, Adam, and SGD.

As shown in Eq. 23, for a Gausian $\mathcal{N}(\boldsymbol{\theta}|\mathbf{m}, \boldsymbol{\Sigma})$, the natural parameter and sufficient statistics are shown below, along with the expectation parameters $\boldsymbol{\mu} = \mathbb{E}_q[\mathbf{T}(\boldsymbol{\theta})]$.

$$\boldsymbol{\lambda} = (\mathbf{Sm}, -\tfrac{1}{2}\mathbf{S}), \quad \mathbf{T}(\boldsymbol{\theta}) = (\boldsymbol{\theta}, \boldsymbol{\theta}\boldsymbol{\theta}^\top), \quad \boldsymbol{\mu} = (\mathbf{m}, \mathbf{mm}^\top + \boldsymbol{\Sigma}),$$

Using these, we can write the natural gradients as gradients with respect to $\boldsymbol{\mu}$, , and then using chain-rule to express them as gradients with respect to $\mathbf{m}$ and $\boldsymbol{\Sigma}$,

$$\tilde{\mathbf{g}}_j(\boldsymbol{\lambda}) = \nabla_{\boldsymbol{\mu}}\mathbb{E}_q[\ell_j(\boldsymbol{\theta})] = \begin{pmatrix} \nabla_{\mathbf{m}}\mathbb{E}_q[\ell_j(\boldsymbol{\theta})] \\ \nabla_{\mathbf{mm}^\top + \boldsymbol{\Sigma}}\mathbb{E}_q[\ell_j(\boldsymbol{\theta})] \end{pmatrix} = \begin{pmatrix} \hat{\mathbf{g}}_j - \hat{\mathbf{H}}_j\mathbf{m} \\ \tfrac{1}{2}\hat{\mathbf{H}}_j, \end{pmatrix}, \tag{27}$$

where in the last equation we define two quantities written in terms of $\nabla\ell_j(\boldsymbol{\theta})$ and $\nabla^2\ell_j(\boldsymbol{\theta})$ by using Price's and Bonnet's theorem [44],

$$\hat{\mathbf{g}}_j = \nabla_{\mathbf{m}}\mathbb{E}_q[\ell_j(\boldsymbol{\theta})] = \mathbb{E}_q[\nabla\ell_j(\boldsymbol{\theta})], \qquad \hat{\mathbf{H}}_j = 2\nabla_{\boldsymbol{\Sigma}}\mathbb{E}_q[\ell_j(\boldsymbol{\theta})] = \mathbb{E}_q[\nabla^2\ell_j(\boldsymbol{\theta})]. \tag{28}$$

Plugging these into the BLR update gives us the following update,

$$\mathbf{S}_t\mathbf{m}_t \leftarrow (1-\rho)\mathbf{S}_{t-1}\mathbf{m}_{t-1} + \rho\sum_{j=0}^{N}\left(\hat{\mathbf{H}}_{j,t-1}\mathbf{m}_{t-1} - \hat{\mathbf{g}}_{j,t-1}\right), \quad \mathbf{S}_t \leftarrow (1-\rho)\mathbf{S}_{t-1} + \rho\sum_{j=0}^{N}\hat{\mathbf{H}}_{j,t-1}$$

where $\hat{\mathbf{g}}_{j,t-1}$ and $\hat{\mathbf{H}}_{j,t-1}$ are quantities similar to before but now evaluated at the $q_{t-1}$. The conjugate model can be written as follows,

$$q_t \propto e^{\boldsymbol{\theta}^\top \mathbf{S}_t\mathbf{m}_t - \tfrac{1}{2}\boldsymbol{\theta}^\top \mathbf{S}_t\boldsymbol{\theta}} \propto (q_{t-1})^{1-\rho}(p_0)^\rho \prod_{j=1}^{N} e^{\boldsymbol{\theta}^\top \hat{\mathbf{i}}_{j,t-1} - \tfrac{1}{2}\boldsymbol{\theta}^\top \hat{\mathbf{I}}_{j,t-1}\boldsymbol{\theta}}$$

The prior above is Gaussian and defined using $q_{t-1}$ and $p_0$. The model uses likelihoods that are Gaussian distribution with information vector $\hat{\mathbf{i}}_{j,t-1} = \rho(\hat{\mathbf{H}}_{j,t-1}\mathbf{m}_{t-1} - \hat{\mathbf{g}}_{j,t-1})$ and information matrix $\hat{\mathbf{I}}_{j,t-1} = \rho\hat{\mathbf{H}}_{j,t-1}$. The likelihood is allowed to be an *improper* distribution, meaning that its integral is not one. This is not a problem as long as $\mathbf{S}_t$ remains positive definite. A valid $\mathbf{S}_t$ can be ensured by either using a Generalized Gauss-Newton approximation to the Hessian [27] or by using the improved BLR of [35]. The former strategy is used in [25] to express BLR iterations as linear models and Gaussian processes. Ultimately, we want to ensure that perturbation in the approximate likelihoods in $q_t$ yields a valid posterior and, as long as this is the case, the conjugate model can be used safely. For instance, in Thm. 4, this issue poses no problem at all.

The BLR update can be rearranged and written in a Newton-like form show below,

$$\text{VON:} \quad \mathbf{m}_t \leftarrow \mathbf{m}_{t-1} - \rho\mathbf{S}_t^{-1}\mathbb{E}_{q_{t-1}}[\nabla\mathcal{L}(\boldsymbol{\theta})], \qquad \mathbf{S}_t \leftarrow (1-\rho)\mathbf{S}_{t-1} + \rho\mathbb{E}_{q_{t-1}}[\nabla^2\mathcal{L}(\boldsymbol{\theta})]. \tag{29}$$

This is called the Variational Online Newton (VON) algorithm. A full derivation is in [27] with details on many of its variants in [28]. The simplest variant is the Online Newton (ON) algorithm, where we use the delta method,

$$\mathbb{E}_{q_t}[\nabla\mathcal{L}(\boldsymbol{\theta})] \approx \nabla\mathcal{L}(\mathbf{m}_t), \qquad \mathbb{E}_{q_t}[\nabla^2\mathcal{L}(\boldsymbol{\theta})] \approx \nabla^2\mathcal{L}(\mathbf{m}_t). \tag{30}$$

Then denoting $\mathbf{m}_t = \boldsymbol{\theta}_t$, we get the following ON update,

$$\text{ON:} \quad \boldsymbol{\theta}_t \leftarrow \boldsymbol{\theta}_{t-1} - \rho\mathbf{S}_t^{-1}\nabla\mathcal{L}(\boldsymbol{\theta}_{t-1}), \qquad \mathbf{S}_t \leftarrow (1-\rho)\mathbf{S}_{t-1} + \rho\nabla^2\mathcal{L}(\boldsymbol{\theta}_{t-1}). \tag{31}$$

To reduce the cost, we can use a diagonal approximation $\mathbf{S}_t = \text{diag}(\mathbf{s}_t)$ where $\mathbf{s}_t$ is a scale vector. Additionally, we can use minibatching to estimate the gradient and hessian (denoted by $\hat{\nabla}$ and $\hat{\nabla}^2$),

$$\text{ON (diagonal+minibatch):} \ \boldsymbol{\theta}_t \leftarrow \boldsymbol{\theta}_{t-1} - \rho\mathbf{s}_t^{-1} \cdot \hat{\nabla}\mathcal{L}(\boldsymbol{\theta}_{t-1}),$$
$$\mathbf{s}_t \leftarrow (1-\rho)\mathbf{s}_{t-1} + \rho\,\text{diag}(\hat{\nabla}^2\mathcal{L}(\boldsymbol{\theta}_{t-1})), \tag{32}$$

where $\cdot$ indicates element-wise product two vectors and $\text{diag}(\cdot)$ extracts the diagonal of a matrix.

Several optimization algorithms can be obtained as special cases from the above variants. For example, to get Newton's method, we set $\rho = 1$ in ON to get

$$\boldsymbol{\theta}_t \leftarrow \boldsymbol{\theta}_{t-1} - [\nabla^2\mathcal{L}(\boldsymbol{\theta}_{t-1})]^{-1}\nabla\mathcal{L}(\boldsymbol{\theta}_{t-1}). \tag{33}$$

RMSprop and Adam can be derived in a similar fashion [28].

In our experiments, we use the improved BLR or iBLR optimizer [35]. We use it to implement an improved version of VON [27, Eqs. 7–8] which ensures that the covariance is always positive-definite, even when the Hessian estimates are not. We use diagonal approximation $\mathbf{S}_t = \text{diag}(\boldsymbol{\sigma}^2)^{-1}$, momentum and minibatching as proposed in [27, 35]. For learning rate $\alpha_t > 0$, momentum $\beta_1, \beta_2 \in [0, 1)$ the iterations are written as follows:

$$
\begin{aligned}
\text{iBLR:} \quad \mathbf{g}_t &\leftarrow \beta_1 \mathbf{g}_{t-1} + (1 - \beta_1)\widehat{\mathbf{g}}_{t-1}, \\
\mathbf{h}_t &\leftarrow \beta_2 \mathbf{h}_{t-1} + (1 - \beta_2)\widehat{\mathbf{h}}_{t-1} + \tfrac{1}{2}(1 - \beta_2)^2(\mathbf{h}_{t-1} - \widehat{\mathbf{h}}_{t-1})^2/(\mathbf{h}_{t-1} + \delta), \\
\mathbf{m}_t &\leftarrow \mathbf{m}_{t-1} - \alpha_t(\mathbf{g}_t + \delta\mathbf{m}_{t-1})/(\mathbf{h}_t + \delta), \\
\boldsymbol{\sigma}_t^2 &\leftarrow 1/(N(\mathbf{h}_t + \delta)).
\end{aligned}
\tag{34}
$$

Here, $\delta > 0$ is the $L_2$-regularization parameter and $\widehat{\mathbf{g}}_{t-1} = \frac{1}{|B|}\sum_{i \in B} \mathbb{E}_{q_{t-1}(\boldsymbol{\theta})}[\nabla \ell_i(\boldsymbol{\theta})]$, $\widehat{\mathbf{h}}_{t-1} = \frac{1}{|B|}\sum_{i \in B} \mathbb{E}_{q_{t-1}(\boldsymbol{\theta})}[\nabla \ell_i(\boldsymbol{\theta})(\boldsymbol{\theta} - \mathbf{m}_{t-1})/\boldsymbol{\sigma}_{t-1}^2]$ denote Monte-Carlo approximations of the expected stochastic gradient and diagonal Hessian under $q_{t-1}(\boldsymbol{\theta}) = \mathcal{N}(\boldsymbol{\theta} \mid \mathbf{m}_{t-1}, \text{diag}(\boldsymbol{\sigma}_{t-1}^2))$ and minibatch $B$. As suggested in [27, 35], we used the reparametrization trick to estimate the diagonal Hessian via gradients only. In practice, we approximate the expectations using a single random sample. We expect multiple samples to further improve the results.

## D  Proof of Thm. 2 and the Beta-Bernoulli Model

From Eq. 27, it directly follows that

$$
\tilde{\mathbf{g}}_j(\boldsymbol{\lambda}) = \nabla_{\boldsymbol{\mu}} \mathbb{E}_q[-\log \tilde{p}_j] = -\nabla_{\boldsymbol{\mu}} \langle \widetilde{\boldsymbol{\lambda}}_j, \mathbb{E}_q[\mathbf{T}(\boldsymbol{\theta})] \rangle = -\nabla_{\boldsymbol{\mu}} \langle \widetilde{\boldsymbol{\lambda}}_j, \boldsymbol{\mu} \rangle = -\widetilde{\boldsymbol{\lambda}}_j.
$$

Using this in Eq. 6, we get the deviation given in Eq. 4.

We will now show an example on Beta-Bernoulli model, which is a conjugate model. We assume the model to be $p(\mathcal{D}, \theta) \propto p(\theta) \prod_i p(y_i|\theta)$ where the prior is $p(\theta) = \text{Beta}(\theta|\alpha_0, \beta_0)$ and likelihoods are $p(y_i|\theta) = \text{Ber}(y_i|\theta)$ with $\mathcal{D}_i = y_i$. This is a conjugate model and the posterior is Beta distribution, that is, it takes the same form as the prior. An expression is given below,

$$
q_* = \text{Beta}(\theta|\alpha_*, \beta_*), \text{ where } \alpha_* = \alpha_0 + \sum_{j=1}^N y_j, \qquad \beta_* = \beta_0 - \sum_{j=1}^N y_j + N.
$$

The posterior for the perturbed dataset $\mathcal{D}^{\backslash i}$ is also available in closed-form:

$$
q_*^{\backslash i} = \text{Beta}(\theta|\alpha_*^{\backslash i}, \beta_*^{\backslash i}), \text{ where } \alpha_*^{\backslash i} = \alpha_0 + \sum_{\substack{j=1, \\ j \neq i}}^N y_j, \qquad \beta_*^{\backslash i} = \beta_0 - \sum_{\substack{j=1, \\ j \neq i}}^N y_j + N - 1.
$$

Therefore the deviations in the posterior parameters can be simply obtained as follows:

$$
\alpha_*^{\backslash i} - \alpha_* = -y_i, \qquad \beta_*^{\backslash i} - \beta_* = y_i - 1
\tag{35}
$$

This result can also be straightforwardly obtained using the MPE. For the Beta distribution $q_{\boldsymbol{\lambda}}(\theta) = \text{Beta}(\theta|\alpha, \beta)$, we have $\boldsymbol{\lambda} = (\alpha - 1, \beta - 1)$, therefore $\boldsymbol{\lambda}_*^{\backslash i} - \boldsymbol{\lambda}_* = (\alpha_*^{\backslash i} - \alpha_*, \beta_*^{\backslash i} - \beta_*)$. For Beta distribution, $\mathbf{T}(\theta) = (\log \theta, \log(1 - \theta))$ and writing the likelihood in an exponential form, we get

$$
p(y_i|\theta) = \text{Ber}(y_i|\theta) \propto \theta^{y_i}(1 - \theta)^{1-y_i} \propto e^{y_i \log \theta + (1-y_i)\log(1-\theta)},
$$

therefore $\widetilde{\boldsymbol{\lambda}}_i = (y_i, y_i - 1)$. Setting $\boldsymbol{\lambda}_*^{\backslash i} - \boldsymbol{\lambda}_* = -\widetilde{\boldsymbol{\lambda}}_i$, we recover the result given in Eq. 35.

## E  Proof of Thm. 3

For linear regression, we have

$$
\nabla \ell_i(\boldsymbol{\theta}) = \mathbf{x}_i(\mathbf{x}_i^\top \boldsymbol{\theta} - y_i), \qquad \nabla^2 \ell_i(\boldsymbol{\theta}) = \mathbf{x}_i \mathbf{x}_i^\top.
$$

Using these in Eq. 8, we get,

$$\tilde{\mathbf{g}}_i(\boldsymbol{\lambda}_*) = \mathbb{E}_q\left[\mathbf{x}_i(\mathbf{x}_i^\top\boldsymbol{\theta} - y_i) - \mathbf{x}_i\mathbf{x}_i^\top\boldsymbol{\theta}_*, \ \tfrac{1}{2}\mathbf{x}_i\mathbf{x}_i^\top\right] = \left(-\mathbf{x}_i y_i, \ \tfrac{1}{2}\mathbf{x}_i\mathbf{x}_i^\top\right),$$

The natural parameter is $\boldsymbol{\lambda}_* = (\mathbf{H}_*\boldsymbol{\theta}_*, -\tfrac{1}{2}\mathbf{H}_*)$. In a similar way, we can define $q_*^{\backslash i}$ and its natural parameter. Using these, we can write Eq. 6 as

$$\mathbf{H}_*^{\backslash i}\boldsymbol{\theta}_*^{\backslash i} - \mathbf{H}_*\boldsymbol{\theta}_* = -\mathbf{x}_i y_i, \qquad\qquad -\tfrac{1}{2}\mathbf{H}_*^{\backslash i} + \tfrac{1}{2}\mathbf{H}_* = \tfrac{1}{2}\mathbf{x}_i\mathbf{x}_i^\top.$$

Substituting the second equation into the first one, we get the first equation below,

$$\mathbf{H}_*^{\backslash i}\boldsymbol{\theta}_*^{\backslash i} - (\mathbf{H}_*^{\backslash i} + \mathbf{x}_i\mathbf{x}_i^\top)\boldsymbol{\theta}_* = -\mathbf{x}_i y_i \quad \Longrightarrow \quad \boldsymbol{\theta}_*^{\backslash i} - \boldsymbol{\theta}_* = (\mathbf{H}_*^{\backslash i})^{-1}\mathbf{x}_i(\mathbf{x}_i^\top\boldsymbol{\theta}_* - y_i) = (\mathbf{H}_*^{\backslash i})^{-1}\mathbf{x}_i e_i.$$

The last equality is exactly Eq. 2. Since linear regression is a conjugate model, an alternate derivation would be to directly use the parameterization $\boldsymbol{\lambda}_j$ of $\tilde{p}_i$ (derived in App. B) and plug it in Thm. 2.

## F  Proof of Thm. 4

For simplicity, we denote

$$\partial\hat{\boldsymbol{\lambda}}_*^{\epsilon_i=0} = \left.\frac{\partial\hat{\boldsymbol{\lambda}}_*^{\epsilon_i}}{\partial\epsilon_i}\right|_{\epsilon_i=0},$$

with $\hat{\boldsymbol{\lambda}}_*^{\epsilon_i}$ as defined in Eq. 7 in the main text. For Gaussian distributions, the natural parameter comes in a pair $\hat{\boldsymbol{\lambda}}_*^{\epsilon_i} = (\mathbf{H}_*^{\epsilon_i}\boldsymbol{\theta}_*^{\epsilon_i}, -\tfrac{1}{2}\mathbf{H}_*^{\epsilon_i})$. Its derivative with respect to $\epsilon_i$ at $\epsilon_i = 0$ can be written as the following by using the chain rule:

$$\partial\hat{\boldsymbol{\lambda}}_*^{\epsilon_i=0} = \left(\mathbf{H}_*\partial\boldsymbol{\theta}_*^{\epsilon_i=0} + \partial\mathbf{H}_*^{\epsilon_i=0}\boldsymbol{\theta}_*, \ -\tfrac{1}{2}\partial\mathbf{H}_*^{\epsilon_i=0}\right).$$

Here, we use the fact that, as $\epsilon_i \to 0$, we have $(\boldsymbol{\theta}_*^{\epsilon_i}, \mathbf{H}_*^{\epsilon_i}) \to (\boldsymbol{\theta}_*, \mathbf{H}_*)$ and also assumed that the limit of the product is equal to the product of the individual limits. Next, we need the expression for the natural gradient, for which we will use Eq. 8 but approximate the expectation by using the delta approximation $\mathbb{E}_{q_*}[g(\boldsymbol{\theta})] \approx g(\boldsymbol{\theta}_*)$ for any function $g$, as shown below to define:

$$\hat{\mathbf{g}}_i(\boldsymbol{\lambda}_*) = \left[\nabla\ell_i(\boldsymbol{\theta}_*) - \nabla^2\ell_i(\boldsymbol{\theta}_*)\boldsymbol{\theta}_*, \ \tfrac{1}{2}\nabla^2\ell_i(\boldsymbol{\theta}_*)\right]$$

The claim is that if we set the perturbed $\partial\hat{\boldsymbol{\lambda}}_*^{\epsilon_i=0} = \hat{\mathbf{g}}_i(\boldsymbol{\lambda}_*)$ we recover Eq. 3, that is, we set

$$\mathbf{H}_*\partial\boldsymbol{\theta}_*^{\epsilon_i=0} + \partial\mathbf{H}_*^{\epsilon_i=0}\boldsymbol{\theta}_* = \nabla\ell(\boldsymbol{\theta}_*) - \nabla^2\ell_i(\boldsymbol{\theta}_*)\boldsymbol{\theta}_*, \qquad\qquad -\tfrac{1}{2}\partial\mathbf{H}_*^{\epsilon_i=0} = \tfrac{1}{2}\nabla^2\ell_i(\boldsymbol{\theta}_*).$$

Plugging the second equation into the first, the second term cancels and we recover Eq. 3.

## G  Extension to Non-Differentiable Loss function

For non-differentiable cases, we can use Eq. 28 to rewrite the BLR of Eq. 29 as

$$\mathbf{m}_t \leftarrow \mathbf{m}_{t-1} - \rho\mathbf{S}_t^{-1}\nabla_{\mathbf{m}}\mathbb{E}_{q_{t-1}}[\mathcal{L}(\boldsymbol{\theta})], \qquad \mathbf{S}_t \leftarrow (1-\rho)\mathbf{S}_{t-1} + 2\rho\nabla_{\boldsymbol{\Sigma}}\mathbb{E}_{q_{t-1}}[\mathcal{L}(\boldsymbol{\theta})], \qquad (36)$$

where $\boldsymbol{\Sigma} = \mathbf{S}^{-1}$. Essentially, we take derivative outside the expectation instead of inside which is valid because the expectation of a non-differentiable function is still differentiable (under some regularity conditions). The same technique can be applied to Eq. 8 to get

$$\tilde{\mathbf{g}}_i(\boldsymbol{\lambda}) = \left(\nabla_{\mathbf{m}}\mathbb{E}_q[\ell_i] - 2\nabla_{\boldsymbol{\Sigma}}\mathbb{E}_q[\ell_i(\boldsymbol{\theta})]\mathbf{m}, \ \nabla_{\boldsymbol{\Sigma}}\mathbb{E}_q[\ell_i(\boldsymbol{\theta})]\right), \qquad (37)$$

and proceeding in the same fashion we can write: $\hat{\mathbf{m}}_t^{\backslash i} - \mathbf{m}_t = (\hat{\mathbf{S}}_t^{\backslash i})^{-1}\nabla_{\mathbf{m}}\mathbb{E}_{q_t}[\ell_i(\boldsymbol{\theta})]$. This is the extension of Eq. 10 to non-differentiable loss functions.

# H  Sensitivity Measures for Sparse Variational Gaussian Processes

Sparse variational GP (SVGP) methods optimize the following variational objective to find a Gaussian posterior approximation $q(\mathbf{u})$ over function values $\mathbf{u} := (f(\mathbf{z}_1), f(\mathbf{z}_2), \ldots, f(\mathbf{z}_M))$ where $\mathcal{Z} := (\mathbf{z}_1, \mathbf{z}_2, \ldots, \mathbf{z}_M)$ is the set of inducing inputs with $M \ll N$:

$$\underline{\mathcal{L}}(\mathbf{m}, \boldsymbol{\Sigma}, \mathcal{Z}, \boldsymbol{\phi}) := \sum_{i=1}^{N} \mathbb{E}_{q(f_i)} \left[ \log p(y_i|f_i) \right] - \mathbb{D}_{\mathrm{KL}}(q(\mathbf{u}) \,\|\, p(\mathbf{u}))$$

where $p(\mathbf{u}) := \mathcal{N}(\mathbf{u}|\mathbf{0}, \mathbf{K_{uu}})$ is the prior with $\mathbf{K_{uu}}$ as the covariance function $\kappa(\cdot, \cdot')$ evaluated at $\mathcal{Z}$, $q(f_i) = \mathcal{N}(f_i|\mathbf{a}_i^\top \mathbf{m}, \mathbf{a}_i^\top \boldsymbol{\Sigma} \mathbf{a}_i + \sigma_i^2)$ is the posterior marginal of $f_i = f(\mathbf{x}_i)$ with $\mathbf{a}_i := \mathbf{K_{uu}^{-1}}\mathbf{k_{ui}}$ and $\sigma_i^2 := \kappa_{ii} - \mathbf{a}_i^\top \mathbf{K_{uu}}\mathbf{a}_i$ as the noise variance of $f_i$ conditioned on $\mathbf{u}$. The objective is also used to optimize hyperparameters $\boldsymbol{\phi}$ and inducing input set $\mathcal{Z}$.

We can optimize the objective using the BLR for which the resulting update is identical to the variational online-newton (VON) algorithm. We first write the natural gradients,

$$\widetilde{\nabla} \mathbb{E}_{q_t(f_i)}[-\log p(y_i|f_i)] = \left( (e_{it} - \beta_{it}\mathbf{a}_i^\top \mathbf{m}_*)\mathbf{a}_i, \ \tfrac{1}{2}\beta_{it}\mathbf{a}_i\mathbf{a}_i^\top \right). \tag{38}$$

where we define

$$e_{it} = \mathbb{E}_{q_t(f_i)}[-\nabla_{f_i} \log p(y_i|f_i)], \qquad \beta_{it} = \mathbb{E}_{q_t(f_i)}[-\nabla^2_{f_i} \log p(y_i|f_i)]$$

We define $\mathbf{A}$ to be a matrix with $\mathbf{a}_i^\top$ as rows, and $\mathbf{e}_t, \boldsymbol{\beta}_t$ to be vectors of $e_{it}, \beta_{it}$. Using these in the VON update, we simplify as follows:

$$\mathbf{S}_{t+1} = (1-\rho)\mathbf{S}_t + \rho \left[ \mathbf{A}^\top \mathrm{diag}(\boldsymbol{\beta}_t)\mathbf{A} + \mathbf{K_{uu}^{-1}} \right] \tag{39}$$

$$\begin{aligned}
\mathbf{m}_{t+1} &= \mathbf{S}_{t+1}^{-1} \left[ (1-\rho)\mathbf{S}_t\mathbf{m}_t - \rho \left( \mathbf{A}^\top \mathbf{e}_t - \mathbf{A}^\top \mathrm{diag}(\boldsymbol{\beta}_t)\mathbf{A}\mathbf{m}_t \right) \right] \\
&= \mathbf{S}_{t+1}^{-1} \left[ \left( (1-\rho)\mathbf{S}_t + \rho \mathbf{A}^\top \mathrm{diag}(\boldsymbol{\beta}_t)\mathbf{A} \right) \mathbf{m}_t - \rho \mathbf{A}^\top \mathbf{e}_t \right] \\
&= \mathbf{S}_{t+1}^{-1} \left[ \left( \mathbf{S}_{t+1} - \rho \mathbf{K_{uu}^{-1}} \right) \mathbf{m}_t - \rho \mathbf{A}^\top \mathbf{e}_t \right] \\
&= \mathbf{S}_{t+1}^{-1} \left[ \mathbf{S}_{t+1}\mathbf{m}_t - \rho \left( \mathbf{A}^\top \mathbf{e}_t + \mathbf{K_{uu}^{-1}}\mathbf{m}_t \right) \right] \\
&= \mathbf{m}_t - \rho \mathbf{S}_{t+1}^{-1} \left[ \mathbf{A}^\top \mathbf{e}_t + \mathbf{K_{uu}^{-1}}\mathbf{m}_t \right].
\end{aligned} \tag{40}$$

For Gaussian likelihood, the updates in Eqs. 39 and 40 coincide with the method of [18], and for non-Gaussian likelihood they are similar to the natural-gradient method by [45], but we use the specific parameterization of [26]. An alternate update rule in terms of site parameters is given by [1] (see Eqs. 22-24).

We are now ready to write the sensitivity measure essentially substituting the gradient in Eq. 10),

$$\mathbf{S}_t^{-1}\nabla_{\mathbf{m}}\mathbb{E}_{q_t(\mathbf{u})}[-\log p(y_i|f_i)] = \mathbf{S}_t^{-1}\mathbf{a}_i\mathbb{E}_{q_t(f_i)}[-\nabla \log p(y_i|f_i)] = \mathbf{S}_t^{-1}\mathbf{a}_i e_{it} \tag{41}$$

We can also see the bi-linear relationship by considering the deviation in the mean of the posterior marginal $f_i(\mathbf{m}) := \mathbf{a}_i^\top \mathbf{m}$,

$$f_i(\mathbf{m}_t^{\setminus i}) - f_i(\mathbf{m}_t) \approx \mathbf{a}_i^\top (\hat{\mathbf{m}}_t^{\setminus i} - \mathbf{m}_t) = \mathbf{a}_i^\top \boldsymbol{\Sigma}_t \mathbf{a}_i e_{it} = v_{it} e_{it} \tag{42}$$

where $v_{it} = \mathbf{a}_i^\top \boldsymbol{\Sigma}_t \mathbf{a}_i$ is the marginal variance of $f_i$.

# I  Experimental Details

## I.1  Neural network architectures

Below, we describe different neural networks used in our experiments,

**MLP (500, 300):** This is a multilayer perceptron (MLP) with two hidden layers of 500 and 300 neurons and a parameter count of around $546\,000$ (using hyperbolic-tangent activations).

**MLP (32, 16):** This is also an MLP with two hidden layers of 32 and 16 neurons, which accounts for around 26 000 parameters (also using hyperbolic tangent activations).

**LeNet5:** This is a standard convolutional neural network (CNN) architecture with three convolution layers followed by two fully-connected layers, corresponding to around 62 000 parameters.

**CNN:** This network, taken from the DeepOBS suite [46], consists of three convolution layers followed by three fully-connected layers with a parameter count of 895 000.

**ResNet–20:** This network has around 274 000 parameters. We use filter response normalization (FRN) [48] as an alternative to batch normalization.

**MLP for USPS:** For the experiment on binary USPS in Fig. 6(a), we use an MLP with three hidden layers of 30 neurons each and a total of around 10 000 parameters.

## I.2 Details of "Do estimated deviations correlate with the truth?"

In Fig. 2, we train neural network classifiers with a cross-entropy loss to obtain $\boldsymbol{\theta}_*$. Due to the computational demand of per-example retraining, the removed examples are randomly subsampled from the training set. We show results over 1000 examples for MNIST and FMNIST and 100 examples for CIFAR10. In the multiclass setting, the expression yields a per-class sensitivity value. We obtain a scalar value for each example by summing over the absolute values of the per-class sensitivities. For training both the original model $\boldsymbol{\theta}_*$ and the perturbed models $\boldsymbol{\theta}_*^{\backslash i}$, we use SGD with a momentum parameter of 0.9 and a cosine learning-rate scheduler. To obtain $\boldsymbol{\theta}_*^{\backslash i}$, we retrain a model that is warmstarted at $\boldsymbol{\theta}_*$. Other details regarding the training setup are given in Table 2. For all models, we do not use data augmentation during training. The resulting $\boldsymbol{\theta}_*$ for MNIST, FMNIST, and CIFAR10 have training accuracies of 99.9%, 95.0%, and 99.9%, respectively. The test accuracies for these models are 98.4%, 91.2% and 76.7%.

| Dataset | Model | $B$ | $\delta$ | $E^*$ | $LR^*$ | $LR^*_{\min}$ | $E^{\backslash i}$ | $LR^{\backslash i}$ | $LR^{\backslash i}_{\min}$ |
|---------|-------|-----|----------|-------|--------|--------------|--------------------|---------------------|---------------------------|
| MNIST | MLP (500, 300) | 256 | 100 | 500 | $10^{-2}$ | $10^{-3}$ | 300 | $10^{-3}$ | $10^{-4}$ |
| FMNIST | LeNet5 | 256 | 100 | 300 | $10^{-1}$ | $10^{-3}$ | 200 | $10^{-3}$ | $10^{-4}$ |
| CIFAR10 | CNN | 512 | 250 | 500 | $10^{-2}$ | $10^{-4}$ | 300 | $10^{-4}$ | $10^{-6}$ |

Table 2: Hyperparameters for predicting true sensitivity in Fig. 2. $B$, $E$ and $LR$ denote batch size, training epochs and learning-rates, respectively. The superscripts $^*$ and $^{\backslash i}$ indicate hyperparameters for training on all data and warmstarted leave-one-out retraining, respectively. $LR_{\min}$ is the minimum learning-rate of the cosine scheduler.

**Additional group removal experiments:** We also study how the deviation for removing a group of examples in a set $\mathcal{M}$ can be estimated using a variation of Eq. 14 for the deviation in predictions at convergence. Denoting the vector of $f_i(\boldsymbol{\theta})$ for $i \in \mathcal{M}$ by $\mathbf{f}_{\mathcal{M}}(\boldsymbol{\theta})$, we get

$$\sigma(\mathbf{f}_{\mathcal{M}}(\boldsymbol{\theta}_*^{\backslash \mathcal{M}})) - \sigma(\mathbf{f}_{\mathcal{M}}(\boldsymbol{\theta}_*)) \approx \boldsymbol{\Lambda}(\boldsymbol{\theta}_*)\mathbf{V}_{\mathcal{M}}(\boldsymbol{\theta}_*)\mathbf{e}_{\mathcal{M}}(\boldsymbol{\theta}_*) \approx \sum_{i \in \mathcal{M}} \sigma'(f_{i*})v_{i*}e_{i*}. \tag{43}$$

where $\boldsymbol{\Lambda}(\boldsymbol{\theta}_*)$ is a diagonal matrix containing all $\sigma'(f_{i*})$, $\mathbf{V}_{\mathcal{M}}(\boldsymbol{\theta}_*) = \nabla \mathbf{f}_{\mathcal{M}}(\boldsymbol{\theta}_*)\mathbf{S}_*^{-1}\nabla \mathbf{f}_{\mathcal{M}}(\boldsymbol{\theta}_*)^\top$ is the prediction covariance of size $M \times M$ where $M$ is the number of examples in $\mathcal{M}$, and $\mathbf{e}_{\mathcal{M}}(\boldsymbol{\theta}_*)$ is the vector of prediction errors. The last approximation above is done to avoid building the covariance, where we ignore the off-diagonal entries of $\mathbf{V}_{\mathcal{M}}(\boldsymbol{\theta}_*)$.

In Fig. 6(a) we consider a binary USPS dataset consisting of the classes for the digits 3 and 5. Using $|\mathcal{M}| = 16$, we show the first and second approximations in Eq. 43 both correlate well with the truth obtained by removing a group and retraining the model. In Fig. 6(b) we do the same on MNIST with $|\mathcal{M}| = 64$, where we see similar trends. For the experiment on binary USPS in Fig. 6(a), we train a MLP with three hidden layers with 30 neurons each. The original model $\boldsymbol{\theta}_*$ is trained for 500 epochs with a learning-rate of $10^{-3}$, a batch size of 32 and a $L_2$-regularization parameter $\delta = 5$. It has 100% training accuracy and 94.8% test accuracy. For the leave-group-out retraining to obtain $\boldsymbol{\theta}_*^{\backslash \mathcal{M}}$, we initialize the model at $\boldsymbol{\theta}_*$, use a learning-rate of $10^{-3}$ and train for 1000 epochs. For the MNIST result in Fig. 6(b) we use the MLP (500, 300) model with the same hyperparameters as for

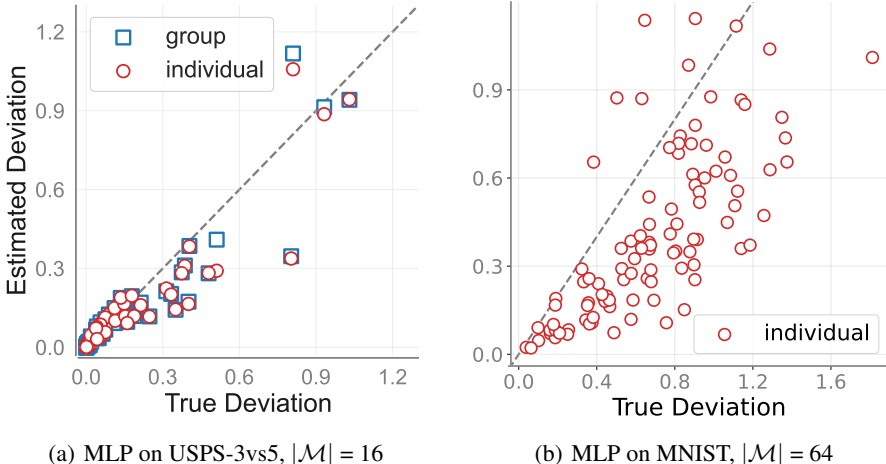

(a) MLP on USPS-3vs5, $|\mathcal{M}| = 16$        (b) MLP on MNIST, $|\mathcal{M}| = 64$

Figure 6: Panel (a) and Panel (b) show that the estimated deviation for removal of groups of examples correlates well with the true deviations obtained by retraining. Each marker corresponds to a removed group of examples. The red circles show the second approximation in Eq. 43. In Panel (a), we additionally show (with blue squares) the first approximation of Eq. 43. We see that the second approximation is quite accurate in this case.

$\boldsymbol{\theta}_*$ in Table 2. For $\boldsymbol{\theta}_*^{\backslash \mathcal{M}}$, we initialize the model at $\boldsymbol{\theta}_*$ and use a cosine schedule of the learning-rate from $10^{-4}$ to $10^{-5}$ over $500$ epochs. We do not use data augmentation. Similarly to the experiments on per-example removal, we use a K-FAC approximation.

| Dataset | Model | $E$ | $LR^*$ | $LR^*_{\min}$ | $LR^{\backslash C}$ | $LR^{\backslash C}_{\min}$ |
|---|---|---|---|---|---|---|
| MNIST | MLP (500, 300) | 500 | $10^{-2}$ | $10^{-3}$ | $10^{-4}$ | $10^{-5}$ |
| MNIST | LeNet5 | 300 | $10^{-1}$ | $10^{-3}$ | $10^{-5}$ | $10^{-6}$ |
| FMNIST | MLP (32, 16) | 300 | $10^{-2}$ | $10^{-3}$ | $10^{-5}$ | $10^{-6}$ |
| FMNIST | LeNet5 | 300 | $10^{-1}$ | $10^{-3}$ | $10^{-4}$ | $10^{-5}$ |

Table 3: Hyperparameters for the class removal experiments in Fig. 3(a) and Fig. 11(d). $B$, $E$ and $LR$ denote batch size, training epochs and learning-rates. The superscripts $*$ and $\backslash C$ indicate hyperparameters for training on all data and warmstarted leave-one-class-out retraining, respectively. $LR_{\min}$ is the minimum learning-rate of the cosine scheduler.

### I.3 Details of "Predicting the effect of class removal on generalization"

For the FMNIST experiment in Fig. 3(a), we use the MLP (32, 16) and LeNet5 models. For the MNIST experiment in Fig. 11(d), we use the MLP (500, 300) and LeNet5 models. The hyperparameters are given in Table 3. The MLP on MNIST has a training accuracy of $99.9\%$ and a test accuracy of $98.4\%$. When using LeNet5, the training and test accuracies are $99.2\%$ and $99.1\%$. On FMNIST, the LeNet5 has an accuracy of $95.0\%$ on the training set, and an accuracy of $91.2\%$ on the test set. On the same dataset, the MLP has a training accuracy of $89.9\%$ and a test accuracy of $86.2\%$. For all models, we use a regularization parameter of $100$ and a batch size of $256$. The leave-one-class-out training is run for $1000$ epochs and the rest of the training setup is same as the previous experiment.

### I.4 Details of "Estimating the leave-one-out cross-validation curves for hyperparameter tuning"

The details of the training setup are in Table 2. Fig. 7 is the same as Fig. 4 but additionally shows the test errors. For visualization purposes, each plot uses a moving average of the plotted lines with a smoothing window. Other training details are similar to previous experiments. All models are

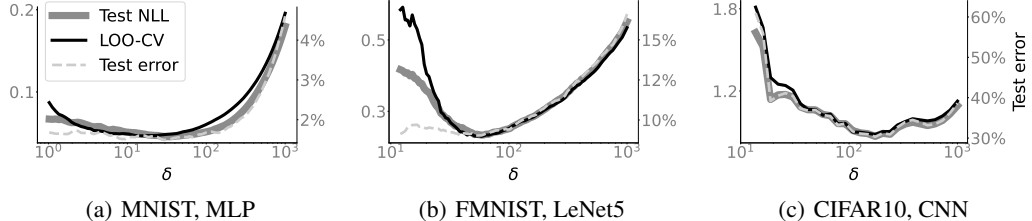

(a) MNIST, MLP       (b) FMNIST, LeNet5       (c) CIFAR10, CNN

Figure 7: Leave-one-out estimation with sensitivities obtained from MPE (Train-LOO-MPE) can accurately estimate the LOO-CV curve for predicting generalization and tuning of the $L_2$-regularization parameter on MNIST, FMNIST and CIFAR-10.

trained from scratch where we use Adam for FMNIST, AdamW [36] for CIFAR10, and SGD with a momentum parameter of 0.9 for MNIST. We use a cosine learning-rate scheduler to anneal the learning-rate. The other hyperparameters are similar to the settings of the models trained on all data from the leave-one-out experiments in Table 2, except for the number of epochs for CIFAR10 where we train for 150 epochs. Similarly to App. I.2, we use a Kronecker-factored Laplace approximation for variance computation and do not employ data augmentation during training.

| Dataset | Model | Number of $\delta$s | Range | Smoothing window |
|---------|-------|---------------------|-------|------------------|
| MNIST | MLP (500, 300) | 96 | $10^0 - 10^3$ | 3 |
| FMNIST | LeNet5 | 96 | $10^1 - 10^3$ | 5 |
| CIFAR10 | CNN | 30 | $10^1 - 10^3$ | 3 |

Table 4: Experimental settings for Fig. 4.

## I.5 Details of "Predicting generalization during the training"

**Details of the training setup:** The experimental details, including test accuracies at the end of training, are listed in Table 5. We use a grid search to determine the regularization parameter $\delta$. The learning-rate is decayed according to a cosine schedule. For diagonal-GGN-LOO and K-FAC-LOO, we use the SGD optimizer with an exception on the FMNIST dataset where we use the AdamW optimizer [36]. In that experiment, we use a weight decay factor of $\delta/N$ replacing the explicit $L_2$-regularization term in the loss in Eq. 1. The regularizer $\mathcal{R}(\boldsymbol{\theta})$ is set to zero. We do not use training data augmentation. For all plots, the LOO-estimate is evaluated periodically during the training, which is indicated with markers.

Additional details on hyperparameters of iBLR are as follows, where $h_0$ is the initialization of the Hessian:

- MNIST, MLP (32, 16): $h_0 = 0.1$
- MNIST, LeNet5: $h_0 = 0.1$
- FMNIST, LeNet5: $h_0 = 0.1$
- CIFAR10, CNN: $h_0 = 0.05$
- CIFAR10, ResNet20: $h_0 = 0.01$

We set $\beta_1 = 0.9$ and $\beta_2 = 0.99999$ in all of those experiments. The magnitude of the prediction variance can depend on $h_0$, which therefore can influence the magnitude of the sensitivities that are perturbing the function outputs in the LOO estimate of Eq. 16. We choose $h_0$ on a grid of four values [0.01, 0.05, 0.1, 0.5] to obtain sensitivities that result in a good prediction of generalization performance.

**Additional Results:** In Fig. 8, we show additional results for MNIST and CIFAR10 that are not included in the main text. For MNIST, we evaluate both on a the MLP (32, 16) model and a LeNet5 architecture. For the additional CIFAR10 results, we use the CNN. In Fig. 9 we include an additional

| Dataset | Model | Method | $LR$ | $LR_{\min}$ | $B$ | $\delta$ | Test acc. |
|---------|-------|--------|------|-------------|-----|----------|-----------|
| MNIST | MLP (32, 16) | iBLR | $10^{-2}$ | $10^{-4}$ | 256 | 80 | 95.6% |
|  |  | diag.-GGN-LOO | $10^{-3}$ | $10^{-4}$ | 256 | 80 | 95.8% |
|  |  | K-FAC-LOO | $10^{-3}$ | $10^{-4}$ | 256 | 80 | 95.8% |
| MNIST | LeNet5 | iBLR | $10^{-2}$ | $10^{-4}$ | 256 | 60 | 97.5% |
|  |  | diag.-GGN-LOO | $10^{-3}$ | $10^{-4}$ | 256 | 60 | 97.4% |
|  |  | K-FAC-LOO | $10^{-3}$ | $10^{-4}$ | 256 | 60 | 97.4% |
| FMNIST | LeNet5 | iBLR | $10^{-1}$ | 0 | 256 | 60 | 90.7% |
|  |  | diag.-GGN-LOO | $10^{-2}$ | $10^{-4}$ | 256 | 60 | 91.0% |
|  |  | K-FAC-LOO | $10^{-2}$ | $10^{-4}$ | 256 | 60 | 91.0% |
| CIFAR10 | CNN | iBLR | $10^{-1}$ | $10^{-4}$ | 512 | 250 | 81.0% |
|  |  | diag.-GGN-LOO | $10^{-1}$ | 0 | 512 | 250 | 75.4% |
|  |  | K-FAC-LOO | $10^{-1}$ | 0 | 512 | 250 | 73.6% |
| CIFAR10 | ResNet–20 | iBLR | $2*10^{-1}$ | 0 | 50 | 10 | 83.4% |

Table 5: Experimental settings for predicting generalization during the training in Fig. 1(b), Fig. 5 and Fig. 8. $B$ and $E$ denote the batch-size and training epochs, respectively. $LR$ and $LR_{\min}$ are the start and end learning-rates of the cosine scheduler. $\delta$ is the regularization parameter. The specification in brackets in the third column indicates the method for computing sensitivities. We use either iBLR or SGD with diagonal GGN (diag.GGN) or K-FAC.

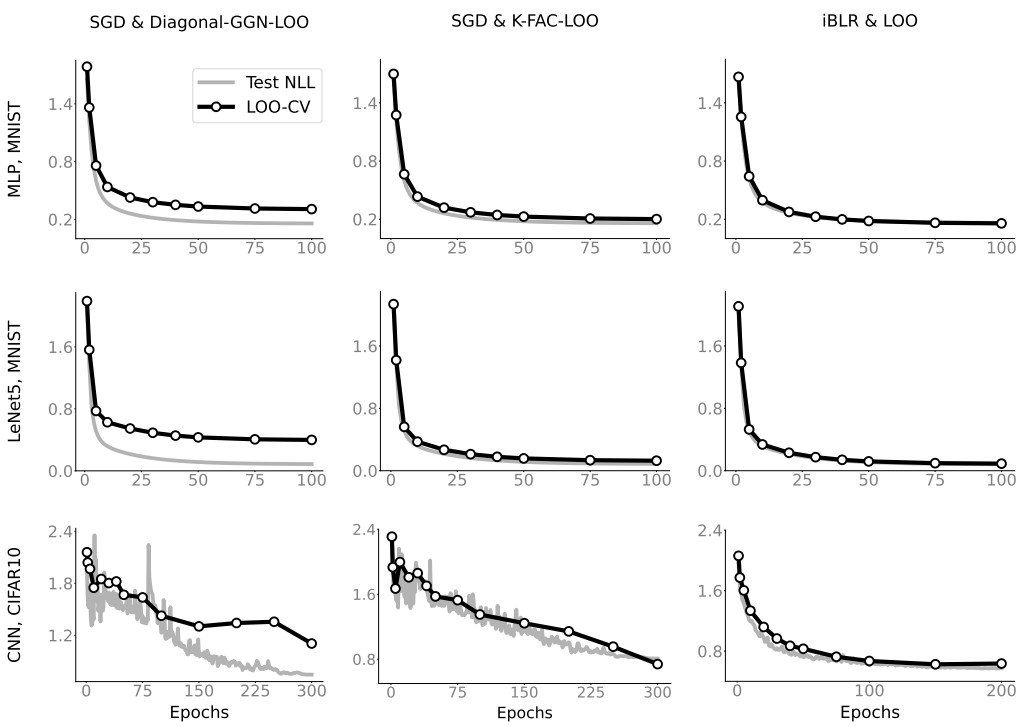

Figure 8: These plots are similar to Fig. 5 but for different model-data pairs. The three rows correspond to MLP on MNIST, LeNet5 on MNIST, and CNN on CIFAR10, respectively. The trends are almost same as those discussed in the main text.

experiment where the model overfits. The K-FAC-LOO estimate deteriorates in this case, but we can still use the LOO as a diagnostic for detecting overfitting and as a stopping criterion. We train

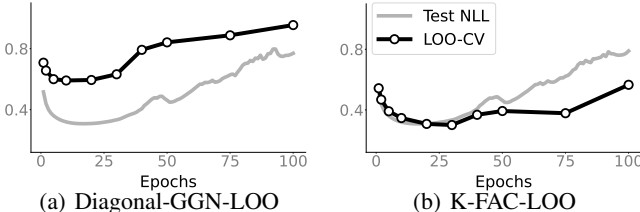

(a) Diagonal-GGN-LOO  (b) K-FAC-LOO

Figure 9: Additional results for training with AdamW where we observe overfitting. We see that K-FAC-LOO deteriorates when the model start to overfit. Both the LOO measures can still be useful tools for diagnosing overfitting. Details of training setup are given in Table 6

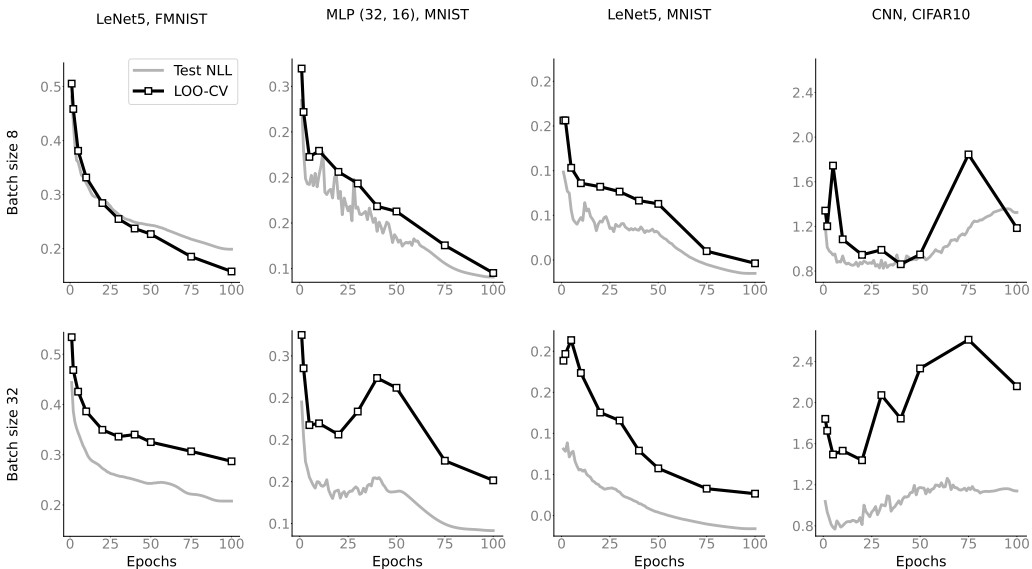

Figure 10: LOO-CV estimates with Adam using the measure suggested in Table 1. The two rows correspond to a batch size of 8 and a batch size of 32, respectively. A smaller batchsize generally decreases the gap between the test NLL and the estimate. Details of the training setup are given in Table 7.

a LeNet5 on FMNIST with AdamW and predict generalization. The trend of the estimated NLL matches the trend of the test NLL in the course of training.

In Fig. 10, we include further results for sensitivity estimation with the Adam optimizer. We use the following update

$$\mathbf{r}_t \leftarrow \beta_1 \mathbf{r}_{t-1} + (1 - \beta_1)\mathbf{g}_t, \quad \mathbf{s}_t \leftarrow \beta_2 \mathbf{s}_{t-1} + (1 - \beta_2)(\mathbf{g}_t \cdot \mathbf{g}_t), \quad \boldsymbol{\theta}_t \leftarrow \boldsymbol{\theta}_{t-1} - \rho\, \mathbf{r}_t / (\sqrt{\hat{\mathbf{s}}_t} + \epsilon),$$

where $\mathbf{g}_t$ is the minibatch gradient, $\beta_1$ and $\beta_2$ are coefficients for the running averages, $\rho$ is a learning-rate, and $\epsilon$ a small damping to stabilize. We construct a diagonal matrix $\mathbf{S}_t = \mathrm{diag}(N\sqrt{\mathbf{s}_t})$ to estimate sensitivity with MPE as suggested in Table 1 ($N$ is the number of training examples). Better results are expected by building better estimates of $\mathbf{S}_t$ as discussed in [27]. As described in section 3.4 of [27], a smaller batch size should improve the estimate, which we also observe in the experiment.

### I.6  Details of "evolution of sensitivities during training"

We use the MPE with iBLR for neural network classification on MNIST, FMNIST and CIFAR10, as well as MPE for logistic regression on MNIST. Experiment details are in Table 8.

For the experiment in Fig. 11(a), we consider Bayesian logistic regression. We set $\delta = 0.1$. The Hessian is always positive-definite due to the convex loss function therefore we use the VON algo-

| Dataset | Model | Method | $LR$ | $LR_{\min}$ | $B$ | $\delta$ | Test acc. |
|---------|-------|--------|------|-------------|-----|----------|-----------|
| FMNIST | LeNet5 | diag., AdamW | $10^{-3}$ | $10^{-3}$ | 256 | 60 | 88.1% |
| | | K-FAC, AdamW | $10^{-3}$ | $10^{-3}$ | 256 | 60 | 87.6% |

Table 6: Experimental settings for predicting generalization during the training in Fig. 9.

| Dataset | Model | $LR$ | $LR_{\min}$ | $\delta$ | Test acc. ($B = 8$) | Test acc. ($B = 32$) |
|---------|-------|------|-------------|----------|---------------------|----------------------|
| MNIST | MLP (32, 16) | $10^{-3}$ | 0 | 80 | 97.3% | 97.4% |
| MNIST | LeNet5 | $10^{-3}$ | 0 | 60 | 99.2% | 99.2% |
| FMNIST | LeNet5 | $10^{-3}$ | 0 | 60 | 91.4% | 91.2% |
| CIFAR10 | CNN | $10^{-3}$ | 0 | 50 | 75.2% | 78.4% |

Table 7: Experimental settings for Fig. 10.

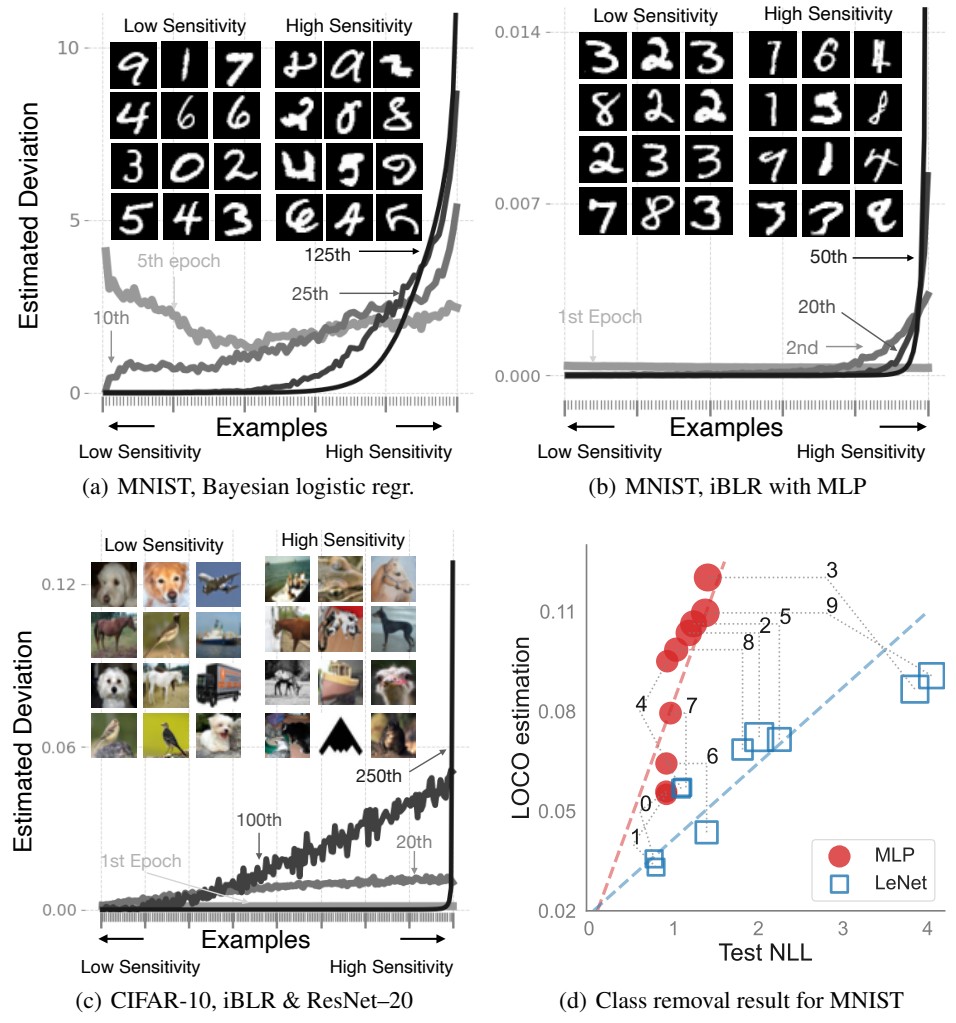

(a) MNIST, Bayesian logistic regr.

(b) MNIST, iBLR with MLP

(c) CIFAR-10, iBLR & ResNet–20

(d) Class removal result for MNIST

Figure 11: Additional experiments similar to Fig. 3(b). In Panel (a), we show the evolution of sensitivities for Bayesian logistic regression on MNIST trained with the VON algorithm. In Panel (b) we use a MLP trained with the iBLR optimizer. In Panel (c), we use a ResNet–20 trained with iBLR on CIFAR-10. Panel (d) shows the class removal result similar to Fig. 3(a), but on MNIST

| Dataset | Model | B | $\delta$ | $E$ |
|---------|-------|---|----------|-----|
| MNIST | MLP (500, 300) | 256 | 30 | 100 |
| FMNIST | LeNet5 | 256 | 60 | 100 |
| CIFAR10 | ResNet–20 | 512 | 35 | 300 |

Table 8: Experimental settings for evolution of sensitivities during training in Fig. 3(b), and Fig. 3.

rithm given in Eq. 29. We use 125 updates with batch-size 200, reaching a test accuracy of around 91% using the mean $\mathbf{m}_t$. We use linear learning-rate decay from 0.005 to 0.001 for the mean $\mathbf{m}$ and a learning-rate of $10^{-5}$ for the precision $\mathbf{S}$. The expectations are approximated using 3 samples drawn from the posterior. We plot sensitivities at iteration $t = 5, 10, 25, 125$. For this example, we use samples from $q_t$ to compute the prediction variance and error (150 samples are used). We sort examples according to their sensitivity at iteration $t = 125$ and then plot their average sensitivities in 60 groups with 100 examples in each group.

For the experiments in Fig. 3(b), Fig. 11(b) and Fig. 11(c), we consider neural network models $\mathbf{f}(\boldsymbol{\theta}_t)$ on FMNIST, MNIST and CIFAR10. We do not use training data augmentation. For CIFAR10 we use a ResNet–20. The expectations in the iBLR are approximated using a single sample drawn from the posterior. For prediction, we use the mean $\mathbf{m}_t$. The test accuracies are 91.3% for FMNIST, 98.5% for MNIST and 80.9% for CIFAR10. We use a cosine learning-rate scheduler with an initial learning-rate of 0.1 and anneal to zero over the course of training. Other experimental details are stated in Table 8. Similar to before, we use sampling to evaluate sensitivity (150 samples are used).

## J  Author Contributions Statement

Authors list: Peter Nickl (PN), Lu Xu (LX), Dharmesh Tailor (DT), Thomas Moellenhoff (TM), Mohammad Emtiyaz Khan (MEK)

All co-authors contributed to developing the main idea. MEK and DT first discussed the idea deriving sensitivity measure based on the BLR. MEK derived the MPE and the results in Sec 3 and DT helped in connecting them to influence functions. PN derived the results in 3.3 and came up with the idea to predict generalization error with LOO-CV. LX adapted it to class-removal. PN wrote the code with help from LX. PN and LX did most of the experiments with some help from TM and regular feedback from everybody. TM did the experiment on the sensitivity evolution during training with some help from PN. All authors were involved in writing and proof-reading of the paper.

## K  Differences Between Camera-Ready Version and Submitted Version

We made several changes to take the feedback of reviewers into account and improve the paper.

1. The writing and organization of the paper were modified to emphasize the generalization to a wide variety of models and algorithms and the applicability of MPE during training.

2. The presentation was changed in Section 3 to emphasize the focus on the conjugate model. Detailed derivations were pushed to the appendices and more focus was put on big picture ideas. Arbitrary perturbations parts were made explicit. Table 1 was added and more focus was put on training algorithms.

3. We added experiments using leave-one-out estimation to predict generalization on unseen test data during traininig. We also added results to study the evolution of sensitivities during training using MPE with iBLR.

