# OpenReview forum: "The Memory-Perturbation Equation: Understanding Model's Sensitivity to Data"
_NeurIPS.cc/2023/Conference — NeurIPS 2023 poster_

### Official Review · Reviewer_fdmt · 2023-07-01

**Soundness:** 3 good
**Presentation:** 2 fair
**Contribution:** 3 good
**Rating:** 6
**Confidence:** 2

**Summary:**

This paper proposes measuring sensitivity to data in some problems in machine learning through what they refer to as the “memory perturbation equation” (MPE), derived using the Bayesian learning rule. They argue that sensitivity properties of machine learning models are not as well understood as they could be in the literature, and therefore propose a remedy to this issue that generalizes some prior approaches. Much of the paper is about justifying the equation and showing that some prior works are special cases. There is also an experimental section that provides some empirical validation of some equations.

**Strengths:**

Although I am not familiar with this general area of research, which seems to be related to influence functions in machine learning, I found the paper interesting and novel. I also found the topic to be highly relevant for NeurIPS and an important one for machine learning in general. The fact that the approach generalizes some prior approaches in a non-trivial way is powerful.

**Weaknesses:**

While I’m generally supportive of this paper, I had a hard time following many of the details in this paper and cannot verify aspects of the approach. This is partly because of my lack of familiarity with the subject, but also because I think the authors have not provided enough background and explanation for a reader with little know-how around related topics. Also, I think the presentation can be improved significantly – please see my detailed questions/comments for some suggestions. I will rely heavily on more knowledgeable reviewers to provide a more thorough assessment.

**Questions:**

Some questions/comments follow:

- For the title, I don’t think “model’s” is the most suitable choice of term. I suggest using either “model” or “models’”.

- “MPE” also stands for most probable explanation in Bayesian modeling, so this could be a point of confusion for some, without context.

- I’m not sure how the word “memory” is suitable for what the authors intend in this work. This is what is written in Section 1: “These highly sensitive examples can be seen as those characterizing the model’s memory; the model is highly sensitive to them and perturbing them can make the model forget its essential knowledge”. Perhaps the authors could explain and justify their choice of term? It may help to connect any other work that they deem to be relevant or inspirational.

- For Fig 1, what is the dataset on which a) and b) panels are based on? Are these all for FMNIST?

- What is the relation between output function f_i(.) and the terms in equation 1? I don’t think this is ever specified. Is f_i the same as l_i?

- Several of the equations (such as those in Section 3) are stated with pointers to prior work but without clear derivation. I understand that there are space limitations, but I had trouble following the correctness of the equations.

- There is a claim in line 207 about how the proposed approach is more principled than some prior work – as someone not familiar with the field, I did not observe much justification for this claim, or perhaps just did not understand it clearly.

- “Sharpness” is mentioned in line 227 but never defined. There are other such terms that don't get enough explanation.

- There are typos, grammatical errors or other such issues in at least the following lines: 39, 41, 59, 61, 63, 68, 110, 113, 124, 161, 221, 236, 244, 249, 268, etc. Note that many of the early typos are due to the incorrect use of verbs for single vs. plural nouns. I suggest the authors carefully review the paper and fix these errors.


**Limitations:**

Limitations are not mentioned in enough detail.

---

> ### Author Rebuttal · Authors · 2023-08-10
>
> >Q1: For the title, I don’t think “model’s” is the most suitable choice of term. I suggest using either “model” or “models’”....“MPE” also stands for most probable explanation in Bayesian modeling, so this could be a point of confusion for some, without context.
>
> A1: Thanks for the suggestions. We will try to avoid the confusion.
>
> >Q2: I’m not sure how the word “memory” is suitable for what the authors intend in this work… perhaps the authors could explain and justify their choice of term? It may help to connect any other work that they deem to be relevant or inspirational.
>
> A2: We use the word “memory” due to a connection to Bayesian methods in psychology where similar concepts are used, e.g., see “Bayesian sets” by Ghahramani and Heller, 2005, where log-ratios of posteriors are used to retrieve “relevant” examples. There is work in psychology (e.g., representativeness) to connect these ideas to human memory. We did not go into detail about this, but hope to write about this connection in a future study.
>
> >Q3: For Fig 1, what is the dataset on which a) and b) panels are based on? Are these all for FMNIST?
>
> A3: Panel b) is for MNIST and c) is for FMNIST, while Panel a) is just an illustration. We will fix the caption to clarify this.
>
> >Q4: What is the relation between output function f_i(.) and the terms in equation 1? I don’t think this is ever specified. Is f_i the same as l_i?
>
> A4: $f_i$ is the model’s output, while $\ell_i$ is the loss function. Line 60 shows an example for linear regression and the mean-squared error loss, where $f_i = x_i^{T} \theta$.
>
> >Q5: Several of the equations (such as those in Section 3) are stated with pointers to prior work but without clear derivation. I understand that there are space limitations, but I had trouble following the correctness of the equations.
>
> A5: For the final version, we add derivations to the appendix to make the work more self-contained.
>
> >Q6: There is a claim in line 207 about how the proposed approach is more principled than some prior work – as someone not familiar with the field, I did not observe much justification for this claim, or perhaps just did not understand it clearly.
>
> A6: We will expand this part. Essentially, the previous work uses an ad-hoc smoothing mechanism to make the loss differentiable, while in our work smoothing is naturally done with the posterior distribution (by using the expectation before taking the derivative as shown in line 206).
>
> >Q7: “Sharpness” is mentioned in line 227 but never defined. There are other such terms that don't get enough explanation.
>
> A7: By sharpness we mean the measures studied in the cited paper by Jiang et al. ([11] in the submitted draft). Sharpness of a local minimum captures the sensitivity of the empirical risk to perturbations in model parameters. We will clarify the confusion for the final version. Please let us know the other terms that cause confusion, so we can add explanations.
>
> >Q8: Limitations are not mentioned in enough detail.
>
> A8: We will add this. The major limitation is that better sensitivity may require better variances which may not always be computationally feasible.

---

> > ### Comment · Reviewer_fdmt · 2023-08-17
> > **Thanks for the response**
> >
> > I thank the authors for their response to my comments. Again, I suggest they edit the paper slightly to make it more readable for a generic reader. I also think more details about terms such as "memory" and "sharpness" are important; hopefully they will make such edits in a revision.

---

> > > ### Author Response · Authors · 2023-08-21
> > >
> > > Thanks for your feedback. We will take it into account to improve this aspect of our work.

---

### Official Review · Reviewer_YYDA · 2023-07-06

**Soundness:** 3 good
**Presentation:** 3 good
**Contribution:** 3 good
**Rating:** 6
**Confidence:** 3

**Summary:**

This paper studies the problem of the model’s sensitivity to its training data (e.g., the counterfactual of how the model's performance will change if trained without certain data), commonly referred to as the "data influence" problem. This paper presents the "memory perturbation equation (MPE)" to study the model’s sensitivity to such perturbations and uses Bayesian learning rule (BLR) approximated using an exponential family distribution. Then, the BLR can be used to estimate the deviation caused by the removal of a group of data.

The paper shows that MPE with Gaussian posterior recovers the celebrated Influence Functions (INF) in both linear and nonlinear cases, and INF can be considered as a special case of the MPE. The work also provides a variety of empirical results on MNIST to CIFAR-10 that shows the estimated sensitivity highly correlates with the actual deviation that would be caused by removing the data and thus can be used as its predictions.

**Strengths:**

This paper is an interesting addition to the research line on "data influence" problems. The paper is well-written. The conceptual development of this paper is clear and smooth. The methodology development and its derivations are solid and the paper also provides insightful remarks supporting the narrative. The method presented in the paper is general and can be potentially applied to a variety of work. Experiment results are nicely presented, clear and interesting.

**Weaknesses:**

I am listing some minor issues that may be improved.

In this work, despite strong correlations to the Influence Function have been shown, the comparison to data influence methods in general isn't comprehensive. There are often works talking about recovering linear influence function or using Taylor approximation to extend to nonlinear cases. It would be nice to see how it relates to or compares to broader works, such as [1] or TracIn [2], just to name a few.
[1] Repairing Neural Networks by Leaving the Right Past Behind. NeurIPS 2022.
[2] Estimating training data influence by tracing gradient descent. NeurIPS 2020.

The empirical results of the proposed approach seem isn't especially strong. As we can see, the correlation between prediction and the actual counterfactual deteriorates quickly as the model becomes bigger. It seems quite well for MNIST/MLP, but less satisfactory for LeNet/FMNIST and CNN/CIFAR-10. I appreciate the honesty of authors that show these results in a straightforward manner such that the reader can directly see the real capability of these methods. Because of that, I would think the relatively weak empirical results do not pose a major compromise to the contribution of this work.

**Questions:**

I was wondering about the approximation error for using exponential family distributions or Gaussian priors. It would be nice if the paper can provide some discussions on this to help better understanding.

What is the computational overhead of the proposed method, in terms of time complexity and memory demand?

**Limitations:**

The empirical results are relatively weak. The computational demand and its scalability are unclear.

---

> ### Author Rebuttal · Authors · 2023-08-10
>
> >Q1: Despite strong correlations to the Influence Function have been shown, the comparison to data influence methods in general isn't comprehensive…. It would be nice to see how it relates to or compares to broader works, such as [1] or TracIn [2]..
>
> A1: This is a good point. The MPE unifies many such approaches (e.g., those mentioned in line 75-77) by essentially choosing an appropriate posterior form and an approximation to the natural-gradient (these same approximations are also made in the BLR). Currently, we only give detailed derivations for linear regression and neural networks, but we will expand and rewrite these parts to explain the unification of other approaches. To summarize:
> - methods that use Hessians (Koh and Liang 2017, Hara et al. 2019) are obtained by using a Gaussian with an unknown “full” covariance; also see the note in line 171.
> - deep learning optimizers are obtained by using a Gaussian with an unknown “diagonal” covariance.
> - methods that use prediction error or the gradient (see the references in line 76) can often be obtained by using a Gaussian with unknown mean but a fixed known covariance.
>
> We will expand the paper to explain these connections in detail. We will also add the references you suggested.
>
> >Q2: The empirical results of the proposed approach seem isn't especially strong. As we can see, the correlation between prediction and the actual counterfactual deteriorates quickly as the model becomes bigger. It seems quite well for MNIST/MLP, but less satisfactory for LeNet/FMNIST and CNN/CIFAR-10. I appreciate the honesty of authors that show these results in a straightforward manner such that the reader can directly see the real capability of these methods. Because of that, I would think the relatively weak empirical results do not pose a major compromise to the contribution of this work.
>
> A2: We stress that our measures can work on bigger models as well; see Fig. IIa in the 1-page PDF where we show on ResNet-20 (270K parameters) a perfect match between true and estimated generalization error. Also, the MLP used for MNIST in Fig. 2a has 545K parameters. We have several such experiments now.
> The reason behind the less satisfactory results in Fig. 2b and 2c is due to a suboptimal training to produce the ground-truth in the x-axis. The ground truth requires retraining with individual examples removed. We will fix this in the next version of the paper. We do not observe these problems in other experiments (e.g., when predicting generalization error as shown in Fig. IIa)
>
> >Q3: I was wondering about the approximation error for using exponential family distributions or Gaussian priors. It would be nice if the paper can provide some discussions on this to help better understand.
>
> A3: We mention this briefly in line 183 but we will expand this. As mentioned in our response to Q1, by making an appropriate exponential-family approximation we can recover different types of measures. So the quality of approximation affects the quality of the sensitivity measure. See the results shown in Fig. I of the 1-page PDF.
>
> >Q4: “What is the computational overhead of the proposed method, in terms of time complexity and memory demand?”
>
> A4: For diagonal Gaussian approximations, we get deep-learning optimizers that estimate weight-space variances for free (see Sec. 4 of the BLR paper [1]). These can be turned into sensitivity estimates with almost no additional overhead, for instance, using a few Jacobian vector products; see the results shown in Fig. I of the 1-page PDF. In general, as mentioned in line 184, the computational overhead depends on the chosen posterior approximations.
>
> [1] M.E. Khan, H. Rue. The Bayesian learning rule. arXiv 2107.04562, 2023.

---

> > ### Comment · Reviewer_YYDA · 2023-08-19
> > **Thanks for the rebuttal**
> >
> > I have read through the reviews as well as the author's response. I appreciate the authors for their dedicated work and thanks for the response to my comments as well as for providing additional results.
> >
> > As has been pointed out by multiple reviewers, the presentation of this paper could be improved for better accessibility to a broader audience.
> >
> > The questions on approximation errors and computational overhead are not directly addressed. The runtime for the experiments conducted in this work or its order of magnitude is unknown. Its practicality cannot be readily assessed. I recommend the authors to better clarify these parts shall the paper be published.
> >
> > I would keep my score as is. I hope the authors compile the new results and additional discussions into the paper or its Appendix.
> >
> > Regards,
> > Reviewer YYDA

---

> > > ### Author Response · Authors · 2023-08-21
> > >
> > > We thank the reviewer for the feedback and we appreciate their comments! We will improve the presentation and add new results and additional discussions to the paper.
> > >
> > > >The questions on approximation errors and computational overhead are not directly addressed.
> > >
> > > In the rebuttal, we show that a better estimate of variance (better approximation quality) gives better results; see Fig. I in the 1-page PDF.
> > >
> > > >The runtime for the experiments conducted in this work or its order of magnitude is unknown. Its practicality cannot be readily assessed.
> > >
> > > The method is practical and scales well to large problems. For example, in Fig. I, we used Adam, which requires less computation than iBLR but gives slightly worse variance estimates. Both require the computation of Jacobian-vector products but can be scaled to large problems as they both use diagonal weight-space covariances. For Fig. 2, 3, and 4a, we analyzed trained models (e.g., to compare to ground truth deviations), so we used KFAC-Laplace which is a bit more computationally expensive but can still scale to large problems.

---

### Official Review · Reviewer_KnYf · 2023-07-09

**Soundness:** 3 good
**Presentation:** 2 fair
**Contribution:** 4 excellent
**Rating:** 6
**Confidence:** 4

**Summary:**

In this paper, the authors introduce the Memory Perturbation Equation (MPE), a generalization of sensitivity measures of models that uses the recently proposed Bayesian Learning Rule (BLR) as a foundation. Thanks to its Bayesian foundation, the MPE can be used for used for non-converged models and for non-differentiable loss functions.

**Strengths:**

The biggest strength of the paper is the writing. This is an *extremely* well written paper that conveys the idea beautifully. It flows extremely well and is very easy to read and follow along. The generality of the MPE is also mind boggling. This is an extremely general method that I think will be pivotal for deep learning. Lastly, the experiments section did a great job verifying the theory of the method and giving a taste of all of the potential uses for it.

**Weaknesses:**

While I think the writing is fantastic, I think a small exposition on the BLR would due wonders, especially because the paper cites specific equations form the BLR paper. To make the paper more readable, a small section in the appendix with details relevant to the paper and its experiments would do wonders.

While I understand the theory of the method, I have a slight practical disconnect between the experiments from this paper and the BLR. Since Adam naturally arises from the BLR, the posterior variances can be constructed easily and, crucially, the posterior covariance would be diagonal, sidestepping messy matrix inversions. But, from reading the appendix section, it seems like the authors are doing the opposite and are instead getting the Hessian of the loss function (in the appendix the authors write: "Variance computation requires inversion of matrices. For large problems, we use a Kronecker factored Laplace variance approximation as implemented in the laplace package[5]"). If this is indeed the case, then I do find this slightly concerning and is a major limitation of the method as computing Hessians are expensive and practically infeasible for modern architectures. Moreover, it doesn't fall in line of the mantra of the paper which is to leverage the BLR to get these sensitivity measure for practically free.

Please let me know I have misunderstood what is going on.

**Questions:**

N/A

**Limitations:**

A limitation section is missing from the paper, which I think is important to have.

---

> ### Author Rebuttal · Authors · 2023-08-10
>
> >Q1: I think a small exposition on the BLR would due wonders, especially because the paper cites specific equations from the BLR paper. To make the paper more readable, a small section in the appendix with details relevant to the paper and its experiments would do wonders.
>
> A1: Thanks for the suggestion. We will add a section on this.
>
> >Q2: I have a slight practical disconnect between the experiments from this paper and the BLR. Since Adam naturally arises from the BLR, the posterior variances can be constructed easily and, crucially, the posterior covariance would be diagonal, sidestepping messy matrix inversions….. it seems like the authors are …..getting the Hessian of the loss function…. I do find this slightly concerning and is a major limitation of the method as computing Hessians are expensive and practically infeasible for modern architectures. Moreover, it doesn't fall in line with the mantra of the paper which is to leverage the BLR to get these sensitivity measure for practically free.
>
> A2: This is a great question and what you said is exactly the idea: the MPE suggests that better estimates of uncertainty give better estimates of sensitivity; see the 1-page PDF on the experiments, where we compare sensitivity at iterations of methods like SGD, Adam, AdamW, iBLR, etc. Previously, we did KFAC only for some experiments where trained networks were used. For iterations, we always use diagonal versions, but we will fix the writing and make the experiments more clear.
>
> >Q3: A limitation section is missing from the paper, which I think is important to have.
>
> A3: We will add a section to the paper.

---

### Official Review · Reviewer_WQKj · 2023-07-18

**Soundness:** 3 good
**Presentation:** 3 good
**Contribution:** 3 good
**Rating:** 6
**Confidence:** 2

**Summary:**

The paper studies the sensitivity of machine learning models to training data. Previously, sensitivity was often studied through empirical investigations which was costly and do not always generalize across models. In this paper, the authors present the memory perturbation equation (MPE), based on the Bayesian learning rule, as a unifying equation to understand sensitivity of generic ML algorithms to training data. The MPE has two features: Sensitivities to examples is estimated by using natural gradients of those examples alone, and examples with larger natural gradients contributes more to the sensitivity. The authors show that the MPE when specialized to Gaussian posteriors, recovers influence functions in linear models and deep learning, and examples with high prediction error and predictions variances are the most influential and the sensitivity is obtained by multiplying the two; The sensitivity can be estimated cheaply whenever natural gradients are cheap to compute; Finally, the authors show empirically that the MPE can be used to accurately estimate generalization performance on image classification tasks.

**Strengths:**

- The paper is well written and easy to follow for non-experts.
- The proposed MPE formulation is well supported by theoretical foundations based on bayesian learning rule. And according to the authors, the proposed MPE has several nice and intuitive theoretical properties for Gaussian posteriors.
- The experiments are well designed to show various aspects of the the properties of the proposed sensitivity measure, which is interesting and clear.

**Weaknesses:**

- I don’t see much discussions on related work or empirical comparisons to state-of-the-art methods
- The example datasets used in the empirical evaluations seem to be mostly smaller datasets like MNIST and CIFAR.

**Questions:**

Out of curiosity, can the proposed MPE or part of the proposed perturbation-based formulation be used to perform machine learning data valuation, i.e., to evaluate the importance of groups of data samples towards a machine learning task (e.g., https://bair.berkeley.edu/blog/2019/12/16/data-worth/)

**Limitations:**

The authors have adequately addressed the limitations in their discussions section. I don’t see any potential negative impact.

---

> ### Author Rebuttal · Authors · 2023-08-10
>
> >Q1: I don’t see much discussions on related work or empirical comparisons to state-of-the-art methods
>
> A1: Thanks, we will improve this point, but could the reviewer mention some related works they would like us to add? For now, we considered influence function as state-of-the-art and we have new experiments comparing influence functions at iterations of SGD, Adam, AdamW, etc. The experiments clearly show the limitation of existing approaches that do not work well during iterations. See Fig. I in the attached 1-page pdf.
>
> >Q2: The example datasets used in the empirical evaluations seem to be mostly smaller datasets like MNIST and CIFAR.
>
> A2: We agree but this is because our focus is on “unifying existing measures”. Our experiments are essentially validating the theory. We will take experiments on the TinyImageNet dataset into consideration. In the future, we plan to work on large-scale models such as GPT-2.
>
> >Q3: “Out of curiosity, can the proposed MPE or part of the proposed perturbation-based formulation be used to perform machine learning data valuation, i.e., to evaluate the importance of groups of data samples towards a machine learning task (e.g., https://bair.berkeley.edu/blog/2019/12/16/data-worth/)”
>
> A3: Absolutely! We hope to consider such applications in the follow-up study, where we use the framework developed here for larger problems.

---

> > ### Comment · Reviewer_WQKj · 2023-08-19
> >
> > Thanks to the authors for your rebuttal! I don’t have further questions.

---

### Official Review · Reviewer_HWf4 · 2023-07-28

**Soundness:** 3 good
**Presentation:** 2 fair
**Contribution:** 3 good
**Rating:** 7
**Confidence:** 2

**Summary:**

The paper presents a generic method, relying on some Bayesian analogy, to perform instance-wise sensitivity analysis of learned models. A generic updating rule and derivative is presented, as well as its different variations on popular models (Neural networks and Gaussian models). One notable use of the proposed sensitivity indices is to estimate the genralisation capabilities of a given model.

Experiments are performed on classical MNIST/FMNIST data sets to confirm that the behaviour of the method is the expected one.

**Strengths:**

+: a versatile method to perform sensitivity analysis

+: a strong technical proposal, at least from what I can assess

**Weaknesses:**

See questions for more detailed comment

-: link with Bayesian approach remains to some extent unclear in the general case.

-: the approach mainly consider instance-wise sensitivity indices, and it is unclear whether this is sufficient in general?

(minor) -: the paper contains a number of typos, and a final read could be useful to correct them (e.g., L41 "can estimated", L63 "the expressions shows", L59, "data as large", L73, "such topics as rare", ...)



**Questions:**

* Bayesian approximation validity: In L92, I found it very strange that one can assume that there is always a proportional link between a Bayesian posterior and the exponential of a Loss function. Sure, the exponential allow to turn the additivity of the loss function into a product form mimicking Bayesian updating, but is it guaranteed that any loss function can be related to a legitimate (proper) prior/likelihood pair in a standard Bayesian framework? Similarly, except for the Gaussian case, what is the quality one can expect from an appriximation using the exponential family? This is a bit hard to figure out, as all examples given in the rest of the paper rest on the (specific) Gaussian case? A good summary of this whole question would be "How Bayesian (in the sense of satiisfying the axioms and coherence of Bayesian approaches, as expressed by, e.g., De Finetti) actually is the BLR rule?"

* In echo to my previous question, do we have an idea of how often the premisses of Theorem 2 ($q_\lambda=p(\theta|\mathcal{D})$) are true? Verifying it seems highly non-trivial to me.

* Thanks to the underlying assumptions made, it makes perfect sense to consider that perturbation effects are additive and that samples have independent effects on the model. However, it would seem more reasonable to consider that, in practice, samples do have interactions between them and that the impact on the learned model are not merely additive, hence the need to include some possible dependencies/interactions between the samples. Maybe the authors could clarify a bit the situations where the additivity assumption makes sense?

* While the removal of an observation is perfectly interpretable from the data perspective, I would appreciate some clarification about the $\epsilon_i$ perturbation. From what I get, this would correspond to weighting the data  in the learning scheme, and see what happens in this case. Is this correct, or would it be more accurate to consider that the data can move within a given neighboorhood within the input space (a perturbation that would probably make more sense from a physical viewpoint, even if I am fine with the weighting interpretation).

* Suggestion: I do not really see the advantage of repeating figures that can be found in some other places in Figure 1, especially as a reader will not have the necessay elements to properly interpret them at this stage. Maybe remove them to better discuss some other points?

**Limitations:**

See questions about some possible limitations, not especially discussed/mentioned in the paper.

---

> ### Author Rebuttal · Authors · 2023-08-10
>
> >Q1: Bayesian approximation validity: In L92, I found it very strange that one can assume that there is always a proportional link between a Bayesian posterior and the exponential of a Loss function… is it guaranteed that any loss function can be related to a legitimate (proper) prior/likelihood pair in a standard Bayesian framework?
>
> A1: Yes, this is the so-called “generalized” Bayesian framework where any arbitrary loss can be used in place of a likelihood. A short description of this can be found in the Sec 1.2 of the BLR paper [1]. It is well-known in the Bayesian community and several papers exist on this topic; see [2-4]. We will add a short description to the paper.
>
> [1] M.E. Khan, H. Rue. The Bayesian learning rule. arXiv 2107.04562, 2023.\
> [2] T. Zhang. Theoretical analysis of a class of randomized regularization methods. COLT, 1999.\
> [3] PAC-Bayesian supervised classification: The thermodynamics of statistical learn-
> ing. institute of mathematical statistics lecture notes—monograph series 56. IMS, 2007.\
> [4] P. G. Bissiri, C. C. Holmes, and S. G. Walker. A general framework for updating belief
> distributions. Journal of the Royal Statistical Society: Series B (Statistical Methodology), 2016
>
> >Q2: Similarly, except for the Gaussian case, what is the quality one can expect from an approximation using the exponential family? This is a bit hard to figure out, as all examples given in the rest of the paper rest on the (specific) Gaussian case? …. do we have an idea of how often the premises of Theorem 2 ($q_{\lambda} (\theta \lvert D$) are true? Verifying it seems highly non-trivial to me.
>
> A2: We will add a non-Gaussian example for the Beta-Bernoulli Mixture model where the posterior is exact. There are plenty of such cases where the posterior is non-Gaussian and exact. There are no issues with the “quality” of the approximations because they merely reveal the implicit approximations made in the previous non-Bayesian approaches. Our work simply figures out the exact form of the approximations to recover these other approaches from the MPE.
>
> >Q3: How Bayesian actually is the BLR rule?
>
> A3: Yes, the BLR is entirely Bayesian. We encourage the reviewer to refer to the BLR paper, where it is shown that the BLR coincides with the Bayes’ rule for well-known conjugate models and, in others, gives the closest approximation to the posterior according to the KL divergence. For Bayesian models, such as in Bayesian deep learning, this is just variational inference. We will add some text to explain this.
>
> >Q4: Thanks to the underlying assumptions made, it makes perfect sense to consider that perturbation effects are additive and that samples have independent effects on the model. However, it would seem more reasonable to consider that, in practice, samples do have interactions between them and that the impact on the learned model are not merely additive, hence the need to include some possible dependencies/interactions between the samples. Maybe the authors could clarify a bit the situations where the additivity assumption makes sense?
>
> A4: Thanks for the question. We are not sure about which “additive assumption” the reviewer is referring to. In Eq. 6 we make an additive assumption in the “natural-parameter space” but this still takes the interactions into account when estimating effects on the predictors. For example, see Eq. 19 where the covariance matrix (in the first approximation) takes care of the interaction and collinearity between the examples (similarly to the influence function). In Fig. 3a, we show that ignoring the interaction still works on some problems, but considering full covariance is better in general as suggested by Eq. 19.
>
> >Q5: While the removal of an observation is perfectly interpretable from the data perspective, I would appreciate some clarification about the perturbation. From what I get, this would correspond to weighting the data in the learning scheme, and see what happens in this case….
>
> A5: Yes, you are right! For example, we can multiply the loss function of the respective data by $\epsilon_i$, perturb the label $y_i$, or some other perturbation of this kind.
>
> >Q6: Suggestion: I do not really see the advantage of repeating figures that can be found in some other places in Fig. 1, especially as a reader will not have the necessary elements to properly interpret them at this stage. Maybe remove them to better discuss some other points?
>
> A6: Thanks, we will think about this, and also improve the caption to make it more understandable.
>
> >Q7: the paper contains a number of typos…
>
> A7: Thanks, we have fixed them now.

---

> > ### Comment · Reviewer_HWf4 · 2023-08-17
> > **Thanks for the clarification + additional elements**
> >
> > Dear authors,
> >
> > Thank you for your various clarifications and pointers linking Bayesian approaches and loss functions, after which I would gladly raise my score.
> >
> > Regarding the additivity, I gavce a look at Equation (19), however I have the feeling that my point still remain after this. To clarify a bit waht I wanted to say: I did not mean additivity and interactivity in the predictive model, but in the way sensitivities with respect to data removal are computed: from what I get, the MPE obtained by removing two pieces of data amounts to the sum of removing each data point individually. This is the additivity assumption I was mentioning. It may be the case that the covariance matrix mentioned do take care of that, but even in this case it is then approximated by a summative form over each removed data point.
> >
> > Best regards

---

> > > ### Author Response · Authors · 2023-08-21
> > >
> > > We are happy to see an increase in the score. Thank you!
> > >
> > > Regarding the “additivity assumption”, this does not cause a problem in the first approximation in Eq. 19 but (as you point out) only in the second approximation where the sum over examples is used. This is why we did the experiments in Fig. 3a and 3b to understand the effect on a real problem. We will improve our writing to avoid the confusion.
> > >
> > > To understand why the assumption does not cause a problem, we give the equation below for linear regression  where the “exact” deviations are indeed additive. That is, the following holds exactly for the new model $\theta_*^{\backslash \mathcal{M}}$ obtained by removing a group of examples $\mathcal{M}$ from $\theta_*^{}$,
> > >
> > > $S_{*}^{\backslash \mathcal{M}}\theta_*^{\backslash \mathcal{M}} - S_*^{} \theta_*^{} = - \sum_{i \in \mathcal{M}} x_i y_i, \qquad -\frac{1}{2} S_*^{\backslash \mathcal{M}} + \frac{1}{2} S_*^{} = \frac{1}{2} \sum_{i \in \mathcal{M}} x_i x_i^T$
> > >
> > > where  $S_*^{\backslash \mathcal{M}}$ denotes the precision for the new model. The expression, when written in terms of the parameters or the function values, has a covariance matrix that correlates examples.

---

### Author Rebuttal · Authors · 2023-08-10

We appreciate the feedback provided by all reviewers. The main weakness seems to be that \
(1) the relevant background and related work can be improved; and\
(2) the experimental results are weak at times and not on large models.

For issue 1, we believe it can easily be fixed and we have given relevant responses for those. For issue 2, we provided additional results in the 1-page PDF. We also include an experiment on ResNet-20 (270K parameters). The experiments support our main message that (1) better estimates of uncertainty give better estimates of sensitivity, and (2) sensitivity can be estimated well during training. We will further improve the writing and presentation of the paper to take the reviewers' suggestions into account.

---

### Decision · Program_Chairs · 2023-09-21

**Decision:**

Accept (poster)

**Comment:**

The paper was reviewed by 5 knowledgeable reviewers, all of whom recommended acceptance of the paper. Initially raised concerns by the reviewers regarding unclarities, related work, and experimental evaluation were clearly addressed in the authors' rebuttal. Hence, in line with the reviewers' recommendations, I am recommending acceptance of the paper. The authors are encouraged to include the newly presented material in the camera-ready version of their paper and, based on the discussions with the reviewers, improve the clarity of their paper in relevant sections.